# Adversarially Trained Weighted Actor-Critic for Safe Offline Reinforcement Learning

**Honghao Wei**
Washington State University
`honghao.wei@wsu.edu`

**Xiyue Peng**
ShanghaiTech University
`pengxy2024@shanghaitech.edu.cn`

**Arnob Ghosh**
New Jersey Institute of Technology
`arnob.ghosh@njit.edu`

**Xin Liu**
ShanghaiTech University
`liuxin7@shanghaitech.edu.cn`

## Abstract

We propose WSAC (Weighted Safe Actor-Critic), a novel algorithm for Safe Offline Reinforcement Learning (RL) under functional approximation, which can robustly optimize policies to improve upon an arbitrary reference policy with limited data coverage. WSAC is designed as a two-player Stackelberg game to optimize a refined objective function. The actor optimizes the policy against two adversarially trained value critics with small importance-weighted Bellman errors, which focus on scenarios where the actor's performance is inferior to the reference policy. In theory, we demonstrate that when the actor employs a no-regret optimization oracle, WSAC achieves a number of guarantees: $(i)$ For the first time in the safe offline RL setting, we establish that WSAC can produce a policy that outperforms **any** reference policy while maintaining the same level of safety, which is critical to designing a safe algorithm for offline RL. $(ii)$ WSAC achieves the optimal statistical convergence rate of $1/\sqrt{N}$ to the reference policy, where $N$ is the size of the offline dataset. $(iii)$ We theoretically show that WSAC guarantees a safe policy improvement across a broad range of hyperparameters that control the degree of pessimism, indicating its practical robustness. Additionally, we offer a practical version of WSAC and compare it with existing state-of-the-art safe offline RL algorithms in several continuous control environments. WSAC outperforms all baselines across a range of tasks, supporting the theoretical results.

## 1 Introduction

Online safe reinforcement learning (RL) has found successful applications in various domains, such as autonomous driving (Isele et al., 2018), recommender systems (Chow et al., 2017), and robotics (Achiam et al., 2017). It enables the learning of safe policies effectively while satisfying certain safety constraints, including collision avoidance, budget adherence, and reliability. However, collecting diverse interaction data can be extremely costly and infeasible in many real-world applications, and this challenge becomes even more critical in scenarios where risky behavior cannot be tolerated. Given the inherently risk-sensitive nature of these safety-related tasks, data collection becomes feasible only when employing behavior policies satisfies all the safety requirements.

To overcome the limitations imposed by interactive data collection, offline RL algorithms are designed to learn a policy from an available dataset collected from historical experiences by some behavior policy, which may differ from the policy we aim to learn. A desirable property of an effective offline algorithm is the assurance of robust policy improvement (RPI), which guarantees that a learned policy is always at least as good as the baseline behavior policies (Fujimoto et al., 2019; Laroche et al., 2019;

38th Conference on Neural Information Processing Systems (NeurIPS 2024).

Kumar et al., 2019; Siegel et al., 2020; Chen et al., 2022a; Zhu et al., 2023; Bhardwaj et al., 2024). We extend the property of RPI to offline safe RL called safe robust policy improvement (SRPI), which indicates the improvement should be *safe* as well. This is particularly important in offline safe RL. For example, in autonomous driving, an expert human driver operates the vehicle to collect a diverse dataset under various road and weather conditions, serving as the behavior policy. This policy is considered both effective and safe, as it demonstrates proficient human driving behavior while adhering to all traffic laws and other safety constraints. Achieving a policy that upholds the SRPI characteristic with such a dataset can significantly mitigate the likelihood of potential collisions and other safety concerns.

In offline RL, we represent the state-action occupancy distribution of policy $\pi$ over the dataset distribution $\mu$ using the ratio $w^\pi = d^\pi / \mu$. A commonly required assumption is that the $\ell_\infty$ concentrability $C^\pi_{\ell_\infty}$ is bounded, which is defined as the infinite norm of $w^\pi$ for **all** policies (Liu et al., 2019; Chen and Jiang, 2019; Wang et al., 2019; Liao et al., 2022; Zhang et al., 2020). A stronger assumption requires a uniform lower bound on $\mu(a|s)$ (Xie and Jiang, 2021). However, such all-policy concentrability assumptions are difficult to satisfy in practice, particularly for offline safe RL, as they essentially require the offline dataset to have good coverage of **all** unsafe state-action pairs. To address the full coverage requirement, other works (Rashidinejad et al., 2021; Zhan et al., 2022; Chen and Jiang, 2022; Xie et al., 2021; Uehara and Sun, 2021) adapt conservative algorithms by employing the principle of pessimism in the face of uncertainty, reducing the assumption to the best covered policy (or optimal policy) concerning $\ell_\infty$ concentrability. Zhu et al. (2023) introduce $\ell_2$ con-

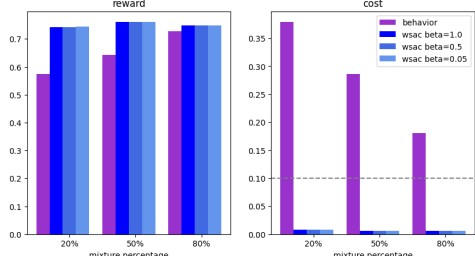

Figure 1: Comparison between WSAC and the behavior policy in the tabular case. The behavior policy is a mixture of the optimal policy and a random policy, with the mixture percentage representing the proportion of the optimal policy. The cost threshold is set to 0.1. We observe that WSAC consistently ensures a safely improved policy across various scenarios, even when the behavior policy is not safe.

centrability to further relax the assumption, indicating that $\ell_\infty$ concentrability is always an upper bound of $\ell_2$ concentrability (see Table 1 for detailed comparisons with previous works). While provable guarantees are obtained using single policy concentrability for unconstrained MDP as Table 1 suggests for the safe RL setting, all the existing studies (Hong et al., 2024; Le et al., 2019) *still* require the coverage on **all** the policies. Further, as Table 1 suggests, the above papers do not guarantee robust safe policy improvement. Our main contributions are summarized below:

1. We prove that our algorithm, which uses average Bellman error, enjoys an optimal statistical rate of $1/\sqrt{N}$ under partial data coverage assumption. *This is the first work that achieves such a result using only single-policy $\ell_2$ concentrability.*

2. We propose a novel offline safe RL algorithm, called Weighted Safe Actor-Critic (WSAC), which can robustly learn policies that improve upon any behavior policy with controlled relative pessimism. We prove that under standard function approximation assumptions, when the actor incorporates a no-regret policy optimization oracle, WSAC outputs a policy that never degrades the performance of a reference policy (including the behavior policy) for a range of hyperparameters (defined later). *This is the first work that provably demonstrates the property of SRPI in offline safe RL setting.*

3. We point out that primal-dual-based approaches Hong et al. (2024) may require **all**-policy concentrability assumption. Thus, unlike, the primal-dual-based appraoch, we propose a novel rectified penalty-based approach to obtain results using **single-policy** concentrability. Thus, we need novel analysis techniques to prove results.

4. Furthermore, we provide a practical implementation of WSAC following a two-timescale actor-critic framework using adversarial frameworks similar to Cheng et al. (2022); Zhu et al. (2023), and test it on several continuous control environments in the offline safe RL benchmark (Liu et al., 2023a). WSAC outperforms all other state-of-the-art baselines, validating the property of a safe policy improvement.

Table 1: Comparison of algorithms for offline RL (safe RL) with function approximation. The parameters $C_{\ell_2}^\pi, C_{\ell_\infty}^\pi, C_{Bellman}^\pi$ refer to different types of concentrabilities, it always hold $C_{\ell_2}^\pi \leq C_{\ell_\infty}^\pi$ and under certain condition $C_{\ell_2}^\pi \leq C_{Bellman}^\pi$, detailed definitions and more discussions can be found in Section 3.3.

| Algorithm | Safe RL | Coverage assumption | Policy Improvement | Suboptimality |
|---|---|---|---|---|
| Xie and Jiang (2021) | No | all policy, $C_{\ell_2}^\pi$ | Yes | $\mathcal{O}(1/\sqrt{N})$ |
| Xie et al. (2021) | No | single-policy, $C_{Bellman}^\pi$ | Yes | $\mathcal{O}(1/\sqrt{N})$ |
| Cheng et al. (2022) | No | single-policy, $C_{Bellman}^\pi$ | Yes & Robust | $\mathcal{O}(1/N^{1/3})$ |
| Ozdaglar et al. (2023) | No | single-policy, $C_{\ell_\infty}^\pi$ | No | $\mathcal{O}(1/\sqrt{N})$ |
| Zhu et al. (2023) | No | single-policy, $C_{\ell_2}^\pi$ | Yes & Robust | $\mathcal{O}(1/\sqrt{N})$ |
| Le et al. (2019) | Yes | all policy, $C_{\ell_\infty}^\pi$ | No | $\mathcal{O}(1/\sqrt{N})$ |
| Hong et al. (2024) | Yes | all policy, $C_{\ell_2}^\pi$ | No | $\mathcal{O}(1/\sqrt{N})$ |
| Ours | Yes | single-policy, $C_{\ell_2}^\pi$ | Yes & Robust | $\mathcal{O}(1/\sqrt{N})$ |

## 2    Related Work

**Offline safe RL:** Deep offline safe RL algorithms (Lee et al., 2022; Liu et al., 2023b; Xu et al., 2022; Chen et al., 2021; Zheng et al., 2024) have shown strong empirical performance but lack theoretical guarantees. To the best of our knowledge, the investigation of policy improvement properties in offline safe RL is relatively rare in the state-of-the-art offline RL literature. Wu et al. (2021) focus on the offline constrained multi-objective Markov Decision Process (CMOMDP) and demonstrate that an optimal policy can be learned when there is sufficient data coverage. However, although they show that CMDP problems can be formulated as CMOMDP problems, they assume a linear kernel CMOMDP in their paper, whereas our consideration extends to a more general function approximation setting. Le et al. (2019) propose a model-based primal-dual-type algorithm with deviation control for offline safe RL in the tabular setting. With prior knowledge of the slackness in Slater's condition and a constant on the concentrability coefficient, an $(\epsilon, \delta)$-PAC error is achievable when the number of data samples $N$ is large enough ($N = \tilde{\mathcal{O}}(1/\epsilon^2)$). These assumptions make the algorithm impractical, and their computational complexity is much higher than ours. Additionally, we consider a more practical, model-free function approximation setting. In another concurrent work (Hong et al., 2024), a primal-dual critic algorithm is proposed for offline-constrained RL settings with general function approximation. However, their algorithm requires $\ell_2$ concentrability for **all** policies, which is not practical as discussed. The reason is that the dual variable optimization in their primal-dual design requires an accurate estimation of all the policies used in each episode, which necessitates coverage over all policies. Moreover, they cannot guarantee the property of SRPI. Moreover, their algorithm requires an additional offline policy evaluation (OPE) oracle for policy evaluation, making the algorithm less efficient.

## 3    Preliminaries

### 3.1    Constrained Markov Decision Process

We consider a Constrained Markov Decision Process (CMDP) $\mathcal{M}$, denoted by $(\mathcal{S}, \mathcal{A}, \mathcal{P}, R, C, \gamma, \rho)$. $\mathcal{S}$ is the state space, $\mathcal{A}$ is the action space, $\mathcal{P} : \mathcal{S} \times \mathcal{A} \to \Delta(\mathcal{S})$ is the transition kernel, where $\Delta(\cdot)$ is a probability simplex, $R : \mathcal{S} \times \mathcal{A} \to [0, 1]$ is the reward function, $C : \mathcal{S} \times \mathcal{A} \to [-1, 1]$ is the cost function, $\gamma \in [0, 1)$ is the discount factor and $\rho : \mathcal{S} \to [0, 1]$ is the initial state distribution. We assume $\mathcal{A}$ is finite while allowing $\mathcal{S}$ to be arbitrarily complex. We use $\pi : \mathcal{S} \to \Delta(\mathcal{A})$ to denote a stationary policy, which specifies a distribution over actions for each state. At each time, the agent

observes a state $s_t \in \mathcal{S}$, takes an action $a_t \in \mathcal{S}$ according to a policy $\pi$, receives a reward $r_t$ and a cost $c_t$, where $r_t = R(s_t, a_t), c_t = C(s_t, a_t)$. Then the CMDP moves to the next state $s_{t+1}$ based on the transition kernel $\mathcal{P}(\cdot|s_t, a_t)$. Given a policy $\pi$, we use $V_r^\pi(s)$ and $V_c^\pi(s)$ to denote the expected discounted return and the expected cumulative discounted cost of $\pi$, starting from state $s$, respectively.

$$V_r^\pi(s) := \mathbb{E}[\sum_{t=0}^{\infty} \gamma^t r_t | s_0 = s, a_t \sim \pi(\cdot|s_t)] \tag{1}$$

$$V_c^\pi(s) := \mathbb{E}[\sum_{t=0}^{\infty} \gamma^t c_t | s_0 = s, a_t \sim \pi(\cdot|s_t)]. \tag{2}$$

Accordingly, we also define the $Q-$value function of a policy $\pi$ for the reward and cost as

$$Q_r^\pi(s,a) := \mathbb{E}[\sum_{t=0}^{\infty} \gamma^t r_t | (s_0, a_0) = (s,a), a_t \sim \pi(\cdot|s_t)] \tag{3}$$

$$Q_c^\pi(s,a) := \mathbb{E}[\sum_{t=0}^{\infty} \gamma^t c_t | (s_0, a_0) = (s,a), a_t \sim \pi(\cdot|s_t)], \tag{4}$$

respectively. As rewards and costs are bounded, we have that $0 \le Q_r^\pi \le \frac{1}{1-\gamma}$, and $-\frac{1}{1-\gamma} \le Q_c^\pi \le \frac{1}{1-\gamma}$. We let $V_{\max} = \frac{1}{1-\gamma}$ to simplify the notation. We further write

$$J_r(\pi) := (1-\gamma)\mathbb{E}_{s\sim\rho}[V_r^\pi(s)], \quad J_c(\pi) := (1-\gamma)\mathbb{E}_{s\sim\rho}[V_c^\pi(s)]$$

to represent the normalized average reward/cost value of policy $\pi$. In addition, we use $d^\pi(s,a)$ to denote the normalized and discounted state-action occupancy measure of the policy $\pi$:

$$d^\pi(s,a) := (1-\gamma)\mathbb{E}[\sum_{t=0}^{\infty} \gamma^t \mathbb{1}(s_t = s, a_t = a)|a_t \sim \pi(\cdot|s_t)],$$

where $\mathbb{1}(\cdot)$ is the indicator function. We also use $d^\pi(s) = \sum_{a \in \mathcal{A}} d^\pi(s,a)$ to denote the discounted state occupancy and we use $\mathbb{E}_\pi$ as a shorthand of $\mathbb{E}_{(s,a)\sim d^\pi}[\cdot]$ or $\mathbb{E}_{s\sim d^\pi}[\cdot]$ to denote the expectation with respect to $d^\pi$. Thus The objective in safe RL for an agent is to find a policy such that

$$\pi \in \arg\max \ J_r(\pi) \qquad \text{s.t. } J_c(\pi) \le 0. \tag{5}$$

*Remark* 3.1. For ease of exposition, this paper exclusively focuses on a single constraint. However, it is readily extendable to accommodate multiple constraints.

## 3.2 Function Approximation

In complex environments, the state space $\mathcal{S}$ is usually very large or even infinite. We assume access to a policy class $\Pi \subseteq (\mathcal{S} \to \Delta(\mathcal{A}))$ consisting of all candidate policies from which we can search for good policies. We also assume access to a value function class $\mathcal{F} \subseteq (\mathcal{S} \times \mathcal{A} \to [0, V_{\max}])$ to model the reward $Q-$functions, and $\mathcal{G} \subseteq (\mathcal{S} \times \mathcal{A} \to [-V_{\max}, V_{\max}])$ to model the cost $Q-$functions of candidate policies. We further assume access to a function class $\mathcal{W} \in \{w : \mathcal{S} \times \mathcal{A} \to [0, B_w]\}$ that represents marginalized importance weights with respect to the offline data distribution. Without loss of generality, we assume that the all-one function is contained in $\mathcal{W}$.

For a given policy $\pi \in \Pi$, we denote $f(s', \pi) := \sum_{a'} \pi(a'|s')f(s', a')$ for any $s \in \mathcal{S}$. The Bellman operator $\mathcal{T}_r^\pi : \mathbb{R}^{\mathcal{S}\times\mathcal{A}} \to \mathbb{R}^{\mathcal{S}\times\mathcal{A}}$ for the reward is defined as

$$(\mathcal{T}_r^\pi f)(s,a) := R(s,a) + \gamma\mathbb{E}_{\mathcal{P}(s'|s,a)}[f(s', \pi)],$$

The Bellman operator $\mathcal{T}_c^\pi : \mathbb{R}^{\mathcal{S}\times\mathcal{A}} \to \mathbb{R}^{\mathcal{S}\times\mathcal{A}}$ for the cost is

$$(\mathcal{T}_c^\pi f)(s,a) := C(s,a) + \gamma\mathbb{E}_{\mathcal{P}(s'|s,a)}[f(s', \pi)].$$

Let $\|\cdot\|_{2,\mu} := \sqrt{\mathbb{E}_\mu[(\cdot)^2]}$ denote the Euclidean norm weighted by distribution $\mu$. We make the following standard assumptions in offline RL setting (Xie et al., 2021; Cheng et al., 2022; Zhu et al., 2023) on the representation power of the function classes:

**Assumption 3.2** (Approximate Realizability). Assume there exists $\epsilon_1 \geq 0$, s.t. for any given policy $\pi \in \Pi$, we have $\min_{f \in \mathcal{F}} \max_{\text{admissible } \nu} \|f - T_r^\pi f\|_{2,\nu}^2 \leq \epsilon_1$, and $\min_{f \in \mathcal{F}} \max_{\text{admissible } \nu} \|f - T_c^\pi f\|_{2,\nu}^2 \leq \epsilon_1$, where $\nu$ is the state-action distribution of any admissible policy such that $\nu \in \{d^\pi, \forall \pi \in \Pi\}$.

Assumption 3.2 assumes that for any policy $\pi \in \Pi$, $Q_r^\pi$ and $Q_c^\pi$ are approximately realizable in $\mathcal{F}$ and $\mathcal{G}$. When $\epsilon_1$ is small for all admissible $\nu$, we have $f_r \approx Q_r^\pi$, and $f_c \approx Q_c^\pi$. In particular, when $\epsilon_1 = 0$, we have $Q_r^\pi \in \mathcal{F}, Q_c^\pi \in \mathcal{F}$ for any policy $\pi \in \Pi$. Note that we do not need Bellman completeness assumption Cheng et al. (2022).

### 3.3 Offline RL

In offline RL, we assume that the available offline data $\mathcal{D} = \{(s_i, a_i, r_i, c_i, s_i')\}_{i=1}^N$ consists of $N$ samples. Samples are i.i.d. (which are common assumptions in unconstrained Cheng et al. (2022), as well as constrained setting Hong et al. (2024)), and the distribution of each tuple $(s, a, r, c, s')$ is specified by a distribution $\mu \in \Delta(\mathcal{S} \times \mathcal{A})$, which is also the discounted visitation probability of a behavior policy (also denoted by $\mu$ for simplicity). In particular, $(s, a) \sim \mu, r = R(s, a), c = C(s, a), s' \sim \mathcal{P}(\cdot|s, a)$. We use $a \sim \mu(\cdot|s)$, to denote that the action is drawn using the behavior policy and $(s, a, s') \sim \mu$ to denote that $(s, a) \sim \mu$, and $s' \sim \mathcal{P}(\cdot|s, a)$.

For a given policy $\pi$, we define the marginalized importance weights $w^\pi(s, a) := \frac{d^\pi(s,a)}{\mu(s,a)}$ which is the ratio between the discounted state-action occupancy of $\pi$ and the data distribution $\mu$. This ratio can be used to measure the concentrability of the data coverage (Xie and Jiang, 2020; Zhan et al., 2022; Rashidinejad et al., 2022; Ozdaglar et al., 2023; Lee et al., 2021).

In this paper we study offline RL with access to a dataset with limited coverage. The coverage of a policy $\pi$ is the dataset can be measured by the weighted $\ell_2$ single policy concentrability coefficient (Zhu et al., 2023; Yin and Wang, 2021; Uehara et al., 2024; Hong et al., 2024):

**Definition 3.3** ($\ell_2$ Concentrability). Given a policy $\pi$, define $C_{\ell_2}^\pi = \|w^\pi\|_{2,\mu} = \|d^\pi/\mu\|_{2,\mu}$.

*Remark* 3.4. The definition here is much weaker than the **all policy** concentrability used in offline RL (Chen and Jiang, 2019) and safe offline RL (Le et al., 2019; Hong et al., 2024), which requires the ratio $\frac{d^\pi(s,a)}{\mu(s,a)}$ to be bounded for all $s \in \mathcal{S}$ and $a \in \mathcal{A}$ and **all** policies $\pi$. In particular, the all-policy concentrability assumption essentially requires the dataset to have full coverage of all policies ((nearly all the state action pairs). This requirement is often violated in practical scenarios. This requirement is even impossible to meet in safe offline RL because it would require collecting data from **every** dangerous state and actions, which clearly is impractical.

In the following lemma, we compare two variants of single-policy concentrability definition with the $\ell_2$ defined in Definition 3.3.

**Lemma 1** (Restate Proposition 2.1 in Zhu et al. (2023)). *Define the $\ell_\infty$ single policy concentrability (Rashidinejad et al., 2021) as $C_{\ell_\infty}^\pi = \|d^\pi/\mu\|_\infty$ and the Bellman-consistent single-policy concentrability (Xie et al., 2021) as $C_{Bellman}^\pi = \max_{f \in \mathcal{F}} \frac{\|f - \mathcal{T}^\pi f\|_{2,d^\pi}^2}{\|f - \mathcal{T}^\pi f\|_{2,\mu}^2}$ ($\mathcal{T}$ could be $\mathcal{T}_r$ or $\mathcal{T}_c$ in our setting) Then, it always holds $(C_{\ell_2}^\pi)^2 \leq C_{\ell_\infty}^\pi, C_{\ell_2}^\pi \leq C_{\ell_\infty}^\pi$ and there exist offline RL instances where $(C_{\ell_2}^\pi)^2 \leq C_{Bellman}^\pi, C_{\ell_2}^\pi \leq C_{Bellman}^\pi$.*

*Remark* 3.5. It is easy to observe that the $\ell_2$ variant is bounded by $\ell_\infty$ and $C_{Bellman}^\pi$ under some cases. There is an example (Example 1) in Zhu et al. (2023) showing that $C_{\ell_2}^\pi$ is bounded by a constant $\sqrt{2}$ while $C_{\ell_\infty}^\pi$ could be arbitrarily large. For the case when the function class $\mathcal{F}$ is highly expressive, $C_{Bellman}^\pi$ could be close to $C_{\ell_\infty}^\pi$ and thus possibly larger than $C_{\ell_2}^\pi$. Intuitively, $C_{\ell_2}^\pi$ implies that only $\mathbb{E}_{d^\pi}[w^\pi(s, a)]$ is bounded, rather, $w^\pi(s, a)$ is bounded for all $(s, a)$ in $\ell_\infty$ concentrability bound.

Given the definition of the concentrability, we make the following assumption on the weight function class $\mathcal{W}$ and a single-policy realizability:

**Assumption 3.6** (Boundedness of $\mathcal{W}$ in $\ell_2$ norm). For all $w \in \mathcal{W}$, assume that $\|w\|_{2,\mu} \leq C_{\ell_2}^*$.

**Assumption 3.7** (Single-policy realizability of $w^\pi$). For some policy $\pi$ that we would like to compete with, assume that $w^\pi \in \mathcal{W}$.

In this paper, we want to study the robust policy improvement on any reference policy, then we assume that we are provided a reference policy $\pi_{\text{ref}}$. Note that in many applications (e.g., scheduling,

networking) we indeed have a reference policy. We want that while applying a sophisticated RL policy it should do better and be safe as well. This is one of the main motivations behind this assumption.

**Assumption 3.8** (Reference Policy). We assume access to a reference policy $\pi_{\text{ref}} \in \Pi$, which can be queried at any state.

In many applications such as networking, scheduling, and control problems, there are existing good enough reference policies. In these cases, a robust and safe policy improvement over these reference policies has practical value. If $\pi_{\text{ref}}$ is not provided, we can simply run a behavior cloning on the offline data to extract the behavior policy as $\pi_{\text{ref}}$ accurately, as long as the size of the offline data set is large enough. More discussion can be found in Section C in the Appendix.

## 4 Actor-Critic with Importance Weighted Bellman Error

Our algorithm design builds upon the constrained actor-critic method, in which we iteratively optimize a policy and improve the policy based on the evaluation of reward and cost. Consider the following actor-critic approach for solving the optimization problem (5):

**Actor:** $\hat{\pi}^* \in \arg\max_{\pi \in \Pi} f_r^\pi(s_0, \pi) \quad s.t. \quad f_c^\pi(s_0, \pi) \leq 0$

**Critic:** $f_r^\pi \in \arg\min_{f \in \mathcal{F}} \mathbb{E}_\mu[((f - \mathcal{T}_r f)(s, a))^2], \quad f_c^\pi \in \arg\min_{f \in \mathcal{G}} \mathbb{E}_\mu[((f - \mathcal{T}_c f)(s, a))^2],$

where we assume that $s_0$ is a fixed initial state, and $f_r(s, \pi) = \sum_{a \in \mathcal{A}} \pi(a|s) f_r(s, a), f_c(s, \pi) = \sum_{a \in \mathcal{A}} \pi(a|s) f_c(s, a)$. The policy is optimized by maximizing the reward $q$ function $f_r$ while ensuring that $f_c$ satisfies the constraint, and the two functions are trained by minimizing the Bellman error. However, this formulation has several disadvantages. 1) It cannot handle insufficient data coverage, which may fail to provide an accurate estimation of the policy for unseen states and actions. 2)It cannot guarantee robust policy improvement. 3) The actor training step is computationally intractable especially when the policy space is extremely large.

To address the insufficient data coverage issue, as mentioned in Xie et al. (2021) the critic can include a Bellman-consistent pessimistic evaluation of $\pi$, which selects the most pessimistic function that approximately satisfies the Bellman equation, which is called absolute pessimism. Then later as indicated by Cheng et al. (2022), instead of using an absolute pessimism, a relative pessimism approach by considering competing to the behavior policy can obtain a robust improvement over the behavior policy. However, this kind of approach can only achieve a suboptimal statistical rate of $N^{1/3}$, and fails to achieve the optimal statistical rate of $1/\sqrt{N}$, then later a weighted average Bellman error (Uehara et al., 2020; Xie and Jiang, 2020; Zhu et al., 2023) could be treated as one possible solution for improving the order. We remark here that all the discussions here are for the traditional *unconstrained* offline RL. Regarding safety, *no existing efficient algorithms in safe offline RL have theoretically demonstrated* the property of robust policy improvement with optimal statistical rate.

**Can Primal-dual based approaches achieve result using only single policy coverability?**: The most commonly used approach for addressing safe RL problems is primal-dual optimization (Efroni et al., 2020; Altman, 1999). As shown in current offline safe RL literature (Hong et al., 2024; Le et al., 2019), the policy can be optimized by maximizing a new unconstrained "reward" $Q-$ function $f_r^\pi(s_0, \pi) - \lambda f_c^\pi(s_0, \pi)$ where $\lambda$ is a dual variable. Then, the dual-variable can be tuned by taking gradient descent step. As we discussed in the introduction, all these require **all** policy concentrability which is not practical especially for safe RL. Important question is whether all policy concentrability assumption can be relaxed. Note that primal-dual algorithm relies on solving the min-max problem $\min_\lambda \max_\pi f_r^\pi(s_0, \pi) - \lambda f_c^\pi(s_0, \pi)$. Recent result (Cui and Du, 2022) shows that single policy concentrability assumption is *not* enough for offline min-max game. Hence, we *conjecture* that using the primal-dual method we can not relax the all policy concentrability assumption. Intuitively, the primal-dual based method (Hong et al., 2024) rely on bounding the regret in dual domain $\sum_k (\lambda_k - \lambda^*)(f_c^{\pi_k} - 0)$, hence, all the policies $\{\pi_k\}_{k=1}^K$ encountered throughout the iteration must be supported by the dataset to evaluate the dual value $\lambda^*(f_c^{\pi_k} - 0)$ where $\lambda^*$ is the optimal dual value.

**Our novelty**: In contrast, we propose an aggression-limited objective function $f_r(s_0, \pi) - \lambda \cdot [f_c(s_0, \pi)]_+$ to control aggressive policies, where $\{\cdot\}_+ := \max\{\cdot, 0\}$. The high-level intuition behind this aggression-limited objective function is that by appropriately selecting a $\lambda$ (usually large enough), we penalize all the policies that are not safe. As a result, the policy that maximizes the

objective function is the optimal safe policy. This formulation is fundamentally different from the traditional primal-dual approach as it does not require dual-variable tuning, and thus, does not require all policy concentrability. In particular, we only need to bound the primal domain regret which can be done as long as the reference policy is covered by the dataset similar to the unconstrained setup.

Combining all the previous ideas together provides the design of our main algorithm named WSAC (**W**eighted **S**afe **A**ctor-**C**ritic). In Section 5, we will provide theoretical guarantees of WSAC and discuss its advantages over existing approaches in offline safe RL. WSAC aims to solve the following optimization problem:

$$\hat{\pi}^* \in \arg\max_{\pi \in \Pi} \mathcal{L}_\mu(\pi, f_r^\pi) - \lambda\{\mathcal{L}_\mu(\pi, f_c^\pi)\}_+$$

$$s.t. \quad f_r^\pi \in \arg\min_{f_r \in \mathcal{F}} \mathcal{L}_\mu(\pi, f_r) + \beta\mathcal{E}_\mu(\pi, f_r), \quad f_c^\pi \in \arg\min_{f_c \in \mathcal{G}} -\lambda\mathcal{L}_\mu(\pi, f_c) + \beta\hat{\mathcal{E}}_\mu(\pi, f_c), \tag{6}$$

where $\mathcal{L}_\mu(\pi, f) := \mathbb{E}_\mu[f(s, \pi) - f(s, a)]$, and $\mathcal{E}_\mu(\pi, f) := \max_{w \in \mathcal{W}} |\mathbb{E}_\mu[w(s, a)((f - T_r^\pi f)(s, a))]|, \hat{\mathcal{E}}_\mu(\pi, f) := \max_{w \in \mathcal{W}} |\mathbb{E}_\mu[w(s, a)((f - T_c^\pi f)(s, a))]|$. This formulation can also be treated as a Stackelberg game (Von Stackelberg, 2010) or bilevel optimization problem. We penalize the objective function only when the approximate cost $Q$-function $f_c^\pi$ of the policy $\pi$ is more perilous than the behavior policy ($f_c^\pi(s, \pi) \geq f_c^\pi(s, a)$) forcing our policy to be as safe as the behavior policy. Maximization over $w$ in for training the two critics can ensure that the Bellman error is small when averaged over measure $\mu \cdot w$ for any $w \in \mathcal{W}$, which turns out to be sufficient to control the suboptimality of the learned policy.

In the following theorem, we show that the solution of the optimization problem (6) is not worse than the behavior policy $\mu$ in both performance and safety for any $\beta \geq 0, \lambda > 0$ than the policy $\mu$ under Assumption 3.2 with $\epsilon_1 = 0$.

**Theorem 4.1.** *Assume that Assumption 3.2 holds with $\epsilon_1 = 0$, and the behavior policy $\mu \in \Pi$, then for any $\beta \geq 0, \lambda > 0$ we have $J_r(\hat{\pi}^*) \geq J_r(\mu)$, and $\{J_c(\hat{\pi}^*)\}_+ \leq \{J_c(\mu)\}_+ + \frac{1}{\lambda}$.*

The result in Theorem 4.1 shows that by selecting $\lambda$ large enough, for any $\beta \geq 0$, the solution can achieve better performance than the behavior policy while maintaining safety that is arbitrarily close to that of the behavior policy. The Theorem verifies the design of our framework which has the potential to have a robust safe improvement.

In the next section, we will introduce our main algorithm WSAC and provide its theoretical guarantees.

## 5 Theoretical Analysis of WSAC

### 5.1 Main Algorithm

In this section, we present the theoretical version of our new model-free offline safe RL algorithm WSAC. Since we only have access to a dataset $\mathcal{D}$ instead of the data distribution. WSAC sovles an empirical version of (6):

$$\hat{\pi} \in \arg\max_{\pi \in \Pi} \mathcal{L}_\mathcal{D}(\pi, f_r^\pi) - \lambda\{\mathcal{L}_\mathcal{D}(\pi, f_c^\pi)\}_+$$

$$s.t. \quad f_r^\pi \in \arg\min_{f_r \in \mathcal{F}} \mathcal{L}_\mathcal{D}(\pi, f_r) + \beta\mathcal{E}_\mathcal{D}(\pi, f_r), \quad f_c^\pi \in \arg\min_{f_c \in \mathcal{G}} -\lambda\mathcal{L}_\mathcal{D}(\pi, f_c) + \beta\hat{\mathcal{E}}_\mathcal{D}(\pi, f_c), \tag{7}$$

where

$$\begin{aligned}
\mathcal{L}_\mathcal{D}(\pi, f) :=& \mathbb{E}_\mathcal{D}[f(s, \pi) - f(s, a)] \\
\mathcal{E}_\mathcal{D}(\pi, f) :=& \max_{w \in \mathcal{W}} |\mathbb{E}_\mathcal{D}[w(s, a)(f(s, a) - r - \gamma f(s', \pi))]| \\
\hat{\mathcal{E}}_\mathcal{D}(\pi, f) :=& \max_{w \in \mathcal{W}} |\mathbb{E}_\mathcal{D}[w(s, a)(f(s, a) - c - \gamma f(s', \pi))]|.
\end{aligned} \tag{8}$$

As shown in Algorithm 1, at each iteration, WSAC selects $f_r^k$ maximally pessimistic and $f_c^k$ maximally optimistic for the current policy $\pi_k$ with a weighted regularization on the estimated Bellman error for reward and cost, respectively (Line $4$ and $6$) to address the worse cases within reasonable range. In order to achieve a safe robust policy improvement, the actor then applies a no-regret policy optimization oracle to update the policy $\pi_{k+1}$ by optimizing the aggression-limited objective function compared with the reference policy (Line 7) $f_r^k(s, a) - \lambda\{f_c^k(s, a) - $

---
**Algorithm 1** **W**eighted **S**afe **A**ctor-**C**ritic (WSAC)
---
1: **Input:** Batch data $\mathcal{D}$, coefficient $\beta, \lambda$. Value function classes $\mathcal{F}, \mathcal{G}$, importance weight function class $\mathcal{W}$, Initialize policy $\pi_1$ randomly. Any reference policy $\pi_{\text{ref}}$. No-regret policy optimization oracle **PO** (Definition 5.1).
2: **for** $k = 1, 2, \ldots, K$ **do**
3:  Obtain the reward state-action value function estimation of $\pi_k$:
4:  $f_r^k \leftarrow \arg\min_{f_r \in \mathcal{F}} \mathcal{L}_{\mathcal{D}}(\pi_k, f_r) + \beta \mathcal{E}_{\mathcal{D}}(\pi_k, f_r)$
5:  Obtain the cost state-action value function estimation of $\pi_k$:
6:  $f_c^k \leftarrow \arg\min_{f_c \in \mathcal{G}} -\lambda \mathcal{L}_{\mathcal{D}}(\pi_k, f_c) + \beta \hat{\mathcal{E}}_{\mathcal{D}}(\pi_k, f_c)$
7:  Update policy: $\pi_{k+1} \leftarrow \textbf{PO}(\pi_k, f_r^k(s,a) - \lambda\{f_c^k(s,a) - f_c^k(s, \pi_{\text{ref}})\}_+, \mathcal{D})$. // $\mathcal{L}_{\mathcal{D}}, \mathcal{E}_{\mathcal{D}}, \hat{\mathcal{E}}_{\mathcal{D}}$ are defined in (5.1)
8: **end for**
9: **Output:** $\bar{\pi} = \text{Unif}(\pi_1, \ldots, \pi_K)$.                          // Uniformly mix $\pi_1, \ldots, \pi_K$
---

$f_c^k(s, \pi_{\text{ref}})\}_+$. Our algorithm is very computationally efficient and tractable compared with existing approaches (Hong et al., 2024; Le et al., 2019), since we do not need another inner loop for optimizing the dual variable with an additional online algorithm or offline policy evaluation oracle. The policy improvement process relies on a no-regret policy optimization oracle, a technique commonly employed in offline RL literature (Zhu et al., 2023; Cheng et al., 2022; Hong et al., 2024; Zhu et al., 2023). Extensive literature exists on such methodologies. For instance, approaches like soft policy iteration (Pirotta et al., 2013) and algorithms based on natural policy gradients (Kakade, 2001; Agarwal et al., 2021) can function as effective no-regret policy optimization oracles. We now formally define the oracle:

**Definition 5.1** (No-regret policy optimization oracle). An algorithm **PO** is called a no-regret policy optimization oracle if for any sequence of functions $f^1, \ldots, f^K$ with $f^k : \mathcal{S} \times \mathcal{A} \to [0, V_{\max}], \forall k \in [K]$. The policies $\pi_1, \ldots, \pi_K$ produced by the oracle **PO** satisfy that for any policy $\pi \in \Pi$ :

$$\epsilon_{opt}^{\pi} \triangleq \frac{1}{K} \sum_{k=1}^{K} \mathbb{E}_{\pi}[f^k(s, \pi) - f^k(s, \pi_k)] = o(1) \tag{9}$$

There indeed exist many methods that can serve as the no-regret oracle, for example, the mirror-descent approach (Geist et al., 2019) or the natural policy gradient approach (Kakade, 2001) of the form $\pi_{k+1}(a|s) \propto \pi_k(a|s) \exp(\eta f^k(s, a))$ with $\eta = \sqrt{\frac{\log |\mathcal{A}|}{2V_{\max}^2 K}}$ (Even-Dar et al., 2009; Agarwal et al., 2021). In the following define $\epsilon_{opt}^{\pi}$ as the error generated from the oracle **PO** by considering $f_r^k(s, a) - \lambda\{f_c^k(s, a) - f_c^k(s, \pi)\}_+$ as the sequence of functions in Definition 5.1, then we have the following guarantee.

**Lemma 2.** *Applying a no-regret oracle **PO** for $K$ episodes with $(f_r^k(s, a) - \lambda\{f_c^k(s, a) - f_c^k(s, \pi)\}_+)$ for an arbitrary policy $\pi$, can guarantee*

$$\frac{1}{K} \sum_{k=1}^{K} \mathbb{E}_{\pi}[f_r^k(s, \pi) - f_r^k(s, \pi_k)] \leq \epsilon_{opt}^{\pi} \tag{10}$$

$$\frac{1}{K} \sum_{k=1}^{K} \mathbb{E}_{\pi}[\{f_c^k(s, \pi_k) - f_c^k(s, \pi)\}_+] \leq \epsilon_{opt}^{\pi} + \frac{V_{\max}}{\lambda}. \tag{11}$$

Lemma 2 establishes that the policy outputted by **PO** with considering the aggression-limited "reward" can have a strong guarantee on the performance of both reward and cost when $\lambda$ is large enough., which is comparable with any competitor policy. This requirement is critical to achieving the performance guarantee of our algorithm and the safe and robust policy improvement. The detailed proof is deferred to Appendix B.2 due to page limit.

## 5.2  Theoretical Guarantees

We are now ready to provide the theoretical guarantees of WSAC Algorithm 1. The complete proof is deferred to Appendix B.3.

**Theorem 5.2** (Main Theorem). *Under Assumptions 3.2 and 3.6, let the reference policy $\pi_{ref} \in \Pi$ in Algorithm 1 be any policy satisfying Assumption 3.7, then with probability at least $1 - \delta$,*

$$J_r(\pi_{ref}) - J_r(\bar{\pi}) \leq \mathcal{O}\left(\epsilon_{stat} + C^*_{\ell_2}\sqrt{\epsilon_1}\right) + \epsilon^{\pi_{ref}}_{opt} \tag{12}$$

$$J_c(\bar{\pi}) - J_c(\pi_{ref}) \leq \mathcal{O}\left(\epsilon_{stat} + C^*_{\ell_2}\sqrt{\epsilon_1}\right) + \epsilon^{\pi_{ref}}_{opt} + \frac{V_{\max}}{\lambda}, \tag{13}$$

*where $\epsilon_{stat} := V_{\max}C^*_{\ell_2}\sqrt{\frac{\log(|\mathcal{F}||\Pi||W|/\delta)}{N}} + \frac{V_{\max}B_w \log(|\mathcal{F}||\Pi||W|/\delta)}{N}$, and $\bar{\pi}$ is the policy returned by Algorithm 1 with $\beta > 0$ and $\pi_{ref}$ as input.*

*Remark* 5.3. When $\epsilon_1 = 0$, i.e., no model misspecification, which states that the true value function belongs to the function class being used to approximate it (the function class is right enough), let $\pi_{\text{ref}}$ be the optimal policy, the results in Theorem 5.2 achieve an optimal dependence statistical rate of $\frac{1}{\sqrt{N}}$(for large $k$), which matches the best existing results. Our algorithm is both statistically optimal and computationally efficient with only **single-policy** assumption rather relying much stronger assumptions of **all** policy concentrability Hong et al. (2024); Le et al. (2019). Hence, if the behavior policy or the reference policy is safe, our result indicates that the policy returned by our algorithm will also be safe (nearly). *Such a guarantee was missing in the existing literature.*

*Remark* 5.4. We also do not need a completeness assumption,which requires that for any $f \in \mathcal{F}$ or $\mathcal{G}$ and $\pi \in \Pi$, it approximately holds that $\mathcal{T}_r f \in \mathcal{F}, \mathcal{T}_c f \in \mathcal{F}$ as required in Xie et al. (2021); Chen et al. (2022b). They need this assumption to address over-estimation issues caused by the $\ell_2$ square Bellman error, but our algorithm can get rid of the strong assumption by using a weighted Bellman error which is a simple and unbiased estimator.

*Remark* 5.5. Our algorithm can compete with any reference policy $\pi_{\text{ref}} \in \Pi$ as long as $w^{\pi_{\text{ref}}} = d^{\pi_{\text{ref}}}/\mu$ is contained in $\mathcal{W}$. The importance ratio of the behavior policy is $w^\mu = d^\mu/\mu = \mu/\mu = 1$ which always satisfies this condition, implying that our algorithm can have a safe robust policy improvement (in Theorem 5.6 discussed below).

### 5.3 A Safe Robust Policy Improvement

A Robust policy improvement (RPI)(Cheng et al., 2022; Zhu et al., 2023; Bhardwaj et al., 2024) refers to the property of an offline RL algorithm that the offline algorithm can learn to improve over the behavior policy, using a wide range of hyperparameters. In this paper, we introduce the property of Safe Robust policy improvement (SRPI) such that the offline algorithm can learn to improve over the behavior policy in both return and safety, using a wide range of hyperparameters. In the following Theorem 5.6 we show that as long as the hyperparameter $\beta = o(\sqrt{N})$, our algorithm can, with high probability, produce a policy with vanishing suboptimality compared to the behavior policy.

**Theorem 5.6** (SRPI). *Under Assumptions 3.2 and 3.6, then with probability at least $1 - \delta$,*

$$J_r(\mu) - J_r(\bar{\pi}) \leq \mathcal{O}\left(\epsilon^\pi_{stat} + C^*_{\ell_2}\sqrt{\epsilon_1}\right) + \epsilon^\mu_{opt} \tag{14}$$

$$J_c(\bar{\pi}) - J_c(\mu) \leq \mathcal{O}\left(\epsilon^\pi_{stat} + C^*_{\ell_2}\sqrt{\epsilon_1}\right) + \epsilon^\mu_{opt} + \frac{V_{\max}}{\lambda}, \tag{15}$$

*where $\epsilon_{stat} := V_{\max}C^*_{\ell_2}\sqrt{\frac{\log(|\mathcal{F}||\Pi||W|/\delta)}{N}} + \frac{V_{\max}B_w \log(|\mathcal{F}||\Pi||W|/\delta)}{N}$, and $\bar{\pi}$ is the policy returned by Algorithm 1 with $\beta \geq 0$ and $\mu$ as input.*

The detailed proofs are deferred to Appendix B.4.

## 6 Experiments

### 6.1 WSAC-Practical Implementation

We introduce a deep RL implementation of WSAC in Algorithm 2 (in Appendix), following the key structure of its theoretical version (Algorithm 1). The reward, cost $Q-$functions $f_r, f_c$ and the policy network $\pi$ are all parameterized by neural networks. The critic losses (line 4) $l_{reward}(f_r)$ and

Table 2: The normalized reward and cost of WSAC and other baselines. The Average line shows the average situation in various environments. The cost threshold is 1. Gray: Unsafe agent whose normalized cost is greater than 1. Blue: Safe agent with best performance

| Environment | BC | | Safe-BC | | BCQL | | BEARL | | CPQ | | COptiDICE | | WSAC | |
|---|---|---|---|---|---|---|---|---|---|---|---|---|---|---|
| | Reward ↑ | Cost ↓ | Reward ↑ | Cost ↓ | Reward ↑ | Cost ↓ | Reward ↑ | Cost ↓ | Reward ↑ | Cost ↓ | Reward ↑ | Cost ↓ | Reward ↑ | Cost ↓ |
| BallCircle | 0.70 | 0.95 | 0.61 | 0.49 | 0.73 | 0.82 | 0.80 | 1.23 | 0.62 | 0.76 | 0.71 | 1.13 | 0.75 | 0.27 |
| CarCircle | 0.57 | 1.43 | 0.57 | 0.65 | 0.79 | 1.19 | 0.84 | 1.87 | 0.67 | 0.28 | 0.49 | 1.52 | 0.68 | 0.59 |
| PointButton | 0.26 | 1.75 | 0.12 | 0.69 | 0.36 | 1.76 | 0.32 | 1.71 | 0.43 | 3.10 | 0.15 | 1.92 | 0.13 | 0.67 |
| PointPush | 0.13 | 0.67 | 0.20 | 1.35 | 0.16 | 1.01 | 0.12 | 0.90 | -0.01 | 2.39 | 0.07 | 1.18 | 0.07 | 0.52 |
| Average | 0.42 | 1.20 | 0.38 | 0.80 | 0.51 | 1.12 | 0.52 | 1.43 | 0.36 | 1.63 | 0.36 | 1.44 | 0.41 | 0.51 |

$l_{cost}(f_c)$ are calculated based on the principles of Algorithm 1, on the minibatch dataset. Optimizing the actor aims to achieve a no-regret optimization oracle, we use a gradient based update on the actor loss (line 5) $l_{actor}(\pi)$. In the implementation we use adaptive gradient descent algorithm ADAM (Kingma and Ba, 2015) for updating two critic networks and the actor network. Algorithm follows standard two-timescale first-order algorithms (Fujimoto et al., 2018; Haarnoja et al., 2018) with a fast learning rate $\eta_{fast}$ on update critic networks and a slow learning rate $\eta_{slow}$ for updating the actor.

## 6.2 Simulations

We present a scalable deep RL version of WSAC in Algorithm 2, following the principles of Algorithm 1. We evaluate WSAC and consider Behavior Cloning (BC), safe Behavior Cloning (Safe-BC), Batch-Constrained deep Q-learning with Lagrangian PID (BCQL) Fujimoto et al. (2019); Stooke et al. (2020) , bootstrapping error accumulation reduction with Lagrangian PID (BEARL) Kumar et al. (2019); Stooke et al. (2020), Constraints Penalized Q-learning (CPQ) Xu et al. (2022) and one of the state-of-the-art algorithms, COptiDICE (Lee et al., 2022) as baselines.

We study several representative environments and focus on presenting "BallCircle". In BallCircle, it requires the ball on a circle in a clockwise direction without leaving the safety zone defined by the boundaries as proposed by Achiam et al. (2017). The ball is a spherical-shaped agent which can freely move on the xy-plane. The reward is dense and increases by the car's velocity and by the proximity towards the boundary of the circle. The cost is incurred if the agent leaves the safety zone defined by the boundaries.

We use the offline dataset from Liu et al. (2019), where the corresponding expert policy are used to interact with the environments and collect the data. To better illustrate the results, we normalize the reward and cost. Our simulation results are reported in Table 2, we observe that WSAC can guarantee that all the final agents are safe, which is most critical in safe RL literature. Even in challenging environments such as PointButton, which most baselines fail to learn safe policies. WSAC has the best results in 3 of the environments. Moreover, WSAC outperforms all the baselines in terms of the average performance, demonstrating its ability to learn a safe policy by leveraging an offline dataset. The simulation results verify our theoretical findings. We also compared WSAC with all the baselines in the case where the cost limits are different, WSAC still outperforms all the other baselines and ensures a safe policy. We further include simulations to investigate the contribution of each component of our algorithm, including the weighted Bellman regularizer, the aggression-limited objective, and the no-regret policy optimization which together guarantee the theoretical results. More details and discussions are deferred to the Appendix D due to page limit.

## 7 Conclusion

In this paper, we explore the problem of offline Safe-RL with a single policy data coverage assumption. We propose a novel algorithm, WSAC, which, for the first time, is proven to guarantee the property of safe robust policy improvement. WSAC is able to outperform any reference policy, including the behavior policy, while maintaining the same level of safety across a broad range of hyperparameters. Our simulation results demonstrate that WSAC outperforms existing state-of-the-art offline safe-RL algorithms. Interesting future work includes combining WSAC with online exploration with safety guarantees and extending the approach to multi-agent settings to handle coupled constraints.

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

# Supplementary Material

## A  Auxiliary Lemmas

In the following, we first provide several lemmas which are useful for proving our main results.

**Lemma 3.** *With probability at least $1 - \delta$, for any $f_r \in \mathcal{F}, f_c \in \mathcal{G}, \pi \in \Pi$ and $w \in \mathcal{W}$, we have*

$$\left| |\mathbb{E}_\mu[(f_r - \mathcal{T}_r^\pi f)w]| - \left|\frac{1}{N} \sum_{(s,a,r,s')} w(s,a)(f_r(s,a) - r - \gamma f_r(s',\pi))\right| \right|$$

$$\leq \mathcal{O}\left( V_{\max}\sqrt{\frac{\log(|\mathcal{F}||\Pi||\mathcal{W}|/\delta)}{N}} + \frac{V_{\max}B_w \log(|\mathcal{F}||\Pi||\mathcal{W}|/\delta)}{N} \right) \tag{16}$$

$$\left| |\mathbb{E}_\mu[(f_c - \mathcal{T}_c^\pi f)w]| - \left|\frac{1}{N} \sum_{(s,a,r,s')} w(s,a)(f_c(s,a) - c - \gamma f_c(s',\pi))\right| \right|$$

$$\leq \mathcal{O}\left( V_{\max}\sqrt{\frac{\log(|\mathcal{G}||\Pi||\mathcal{W}|/\delta)}{N}} + \frac{V_{\max}B_w \log(|\mathcal{G}||\Pi||\mathcal{W}|/\delta)}{N} \right) \tag{17}$$

The proofs can be found in Lemma 4 in Zhu et al. (2023).

**Lemma 4.** *With probability at least $1 - 2\delta$, for any $f_r \in \mathcal{F}, f_c \in \mathcal{G}$ and $\pi \in \Pi$, we have*

$$|\mathcal{E}_\mu(\pi, f_r) - \mathcal{E}_\mathcal{D}(\pi, f_r)| \leq \epsilon_{stat} \tag{18}$$

$$|\mathcal{E}_\mu(\pi, f_c) - \mathcal{E}_\mathcal{D}(\pi, f_c)| \leq \epsilon_{stat}, \tag{19}$$

*where $\epsilon_{stat} := V_{\max} C_{\ell_2}^* \sqrt{\frac{\log(|\mathcal{F}||\mathcal{G}||\Pi||W|/\delta)}{N}} + \frac{V_{\max} B_w \log(|\mathcal{F}||\mathcal{G}||\Pi||W|/\delta)}{N}$.*

*Proof.* Condition on the high probability event in Lemma 3, for any $f_r \in \mathcal{F}, f_c \in \mathcal{G}, \pi \in \Pi$, define

$$w_{\pi,f}^* = \arg \max_{w \in \mathcal{W}} \mathcal{E}_\mu(\pi, f_r) = \arg \max_{w \in \mathcal{W}} |\mathbb{E}_\mu[w(s,a)(f_r - \mathcal{T}_r^\pi f_r)(s,a)]|$$

and define

$$\hat{w}_{\pi,f_r} = \arg \max_{w \in \mathcal{W}} \mathcal{E}_\mathcal{D}(\pi, f_r) = \arg \max_{w \in \mathcal{W}} |\frac{1}{N} \sum_{(s,a,r,s')\in\mathcal{D}} w(s,a)(f_r(s,a) - r - \gamma f_r(s',\pi))|.$$

Then we can have

$$\mathcal{T}_\mu(\pi, f_r) - \mathcal{E}_\mathcal{D}(\pi, f_r)$$

$$= |\mathbb{E}_\mu[w_{\pi,f_r}^*(s,a)(f_r - \mathcal{T}_r^\pi f_r)(s,a)]| - \left|\frac{1}{N} \sum_{(s,a,r,s')} \hat{w}_{\pi,f}(s,a)(f_r(s,a) - r - \gamma f_r'(s',\pi))\right|$$

$$= |\mathbb{E}_\mu[w_{\pi,f_r}^*(s,a)(f_r - \mathcal{T}_r^\pi f_r)(s,a)]| - |\mathbb{E}_\mu[\hat{w}_{\pi,f_r}(s,a)(f_r - \mathcal{T}_r^\pi f_r)(s,a)]|$$

$$\quad + |\mathbb{E}_\mu[\hat{w}_{\pi,f_r}(s,a)(f_r - \mathcal{T}_r^\pi f_r)(s,a)]| - \left|\frac{1}{N} \sum_{(s,a,r,s')} \hat{w}_{\pi,f}(s,a)(f_r(s,a) - r - \gamma f_r'(s',\pi))\right|$$

$$\geq 0 - \epsilon_{stat} = -\epsilon_{stat},$$

where the inequality is true by using the definition of $w_{\pi,f_r}^*$ and Lemma 3. Thus

$$\mathcal{E}_\mu(\pi, f_r) - \epsilon_\mathcal{D}(\pi, f_r)$$

$$= |\mathbb{E}_\mu[w_{\pi,f_r}^*(s,a)(f_r - \mathcal{T}_r^\pi f_r)(s,a)]| - \left|\frac{1}{N} \sum_{(s,a,r,s')} w_{\pi,f_r}^*(s,a)(f_r(s,a) - r - \gamma f_r'(s',\pi))\right|$$

$$+ \left|\frac{1}{N} \sum_{(s,a,r,s')} w_{\pi,f_r}^*(s,a)(f_r(s,a) - r - \gamma f_r'(s',\pi))\right|$$

$$- \left|\frac{1}{N} \sum_{(s,a,r,s')} \hat{w}_{\pi,f_r}(s,a)(f_r(s,a) - r - \gamma f_r'(s',\pi))\right|$$

$$\leq \epsilon_{stat}$$

The proof for the case $|\mathcal{E}_\mu(\pi, f_c) - \mathcal{E}_\mathcal{D}(\pi, f_c)| \leq \epsilon_{stat}$ is similar. $\qquad\square$

**Lemma 5.** *(Empirical weighted average Bellman Error) With probability at least $1 - 2\delta$, for any $\pi \in \Pi$, we have*

$$\mathcal{E}_{\mathcal{D}}(\pi, f_r^\pi) \leq C_{\ell_2}^* \sqrt{\epsilon_1} + \epsilon_{stat} \tag{20}$$

$$\mathcal{E}_{\mathcal{D}}(\pi, f_c^\pi) \leq C_{\ell_2}^* \sqrt{\epsilon_1} + \epsilon_{stat}, \tag{21}$$

*where*

$$f_r^\pi := \operatorname*{arg\,min}_{f_r \in \mathcal{F}} \sup_{admissible\ \nu} \|f_r - \mathcal{T}_r^\pi f_r\|_{2,\nu}^2, \forall \pi \in \Pi$$

$$f_c^\pi := \operatorname*{arg\,min}_{f_c \in \mathcal{G}} \sup_{admissible\ \nu} \|f_c - \mathcal{T}_c^\pi f_c\|_{2,\nu}^2, \forall \pi \in \Pi.$$

*Proof.* Condition on the high probability event in Lemma 4, we have

$$\mathcal{E}_\mu(\pi, f_r^\pi) = \max_{w \in \mathcal{W}} |\mathbb{E}_\mu[w(s,a)((f - T_r^\pi f_r^\pi)(s,a))]|$$

$$\leq \mathcal{E}_\mu(\pi, f_r^\pi) = \max_{w \in \mathcal{W}} |\|w\|_{2,\mu} \|f_\pi - T_r^\pi f)(s,a))]|_{2,\mu}$$

$$\leq C_{\ell_2^*} \sqrt{\epsilon_1},$$

where the first inequality is true because of Cauchy-Schwarz inequality and the second inequality comes from the definition of $f_r^\pi$ and Assumption 3.2, thus we can obtain

$$\mathcal{E}_{\mathcal{D}}(\pi, f_r^\pi) \leq \mathcal{E}_\mu(\pi, f_r^\pi) + \epsilon_{stat} \leq C_{\ell_2}^* \sqrt{\epsilon_1} + \epsilon_{stat}. \tag{22}$$

Following a similar proof we can have

$$\hat{\mathcal{E}}_{\mathcal{D}}(\pi, f_c^\pi) \leq \mathcal{E}_\mu(\pi, f_c^\pi) + \epsilon_{stat} \leq C_{\ell_2}^* \sqrt{\epsilon_1} + \epsilon_{stat}. \tag{23}$$

$\square$

**Lemma 6.** *(Performance difference decomposition, restate of Lemma* 12 *in Cheng et al. (2022)) For an arbitrary policy $\pi, \hat{\pi} \in \Pi$, and $f$ be an arbitrary function over $\mathcal{S} \times \mathcal{A}$. Then we have,*

$$J_\diamond(\pi) - J_\diamond(\hat{\pi})$$
$$= \mathbb{E}_\mu\left[\left(f - \mathcal{T}_\diamond^{\hat{\pi}}\right)(s,a)\right] + \mathbb{E}_\pi\left[\left(\mathcal{T}_\diamond^{\hat{\pi}} f - f\right)(s,a)\right] + \mathbb{E}_\pi[f(s,\pi) - f(s,\hat{\pi})] + \mathcal{L}_\mu(\hat{\pi}, f) - \mathcal{L}_\mu(\hat{\pi}, Q_\diamond^{\hat{\pi}}), \tag{24}$$

*where $\diamond := r$ or $c$.*

*Proof.* We prove the case when $\diamond := r$, the other case is identical. Let $R^{f,\hat{\pi}}(s,a) := f(s,a) - \gamma \mathbb{E}_{s'|(s,a)}[f(s',\hat{\pi})]$ be a virtual reward function for given $f$ and $\hat{\pi}$. According to performance difference lemma (Kakade and Langford, 2002), We first have that

$$(J_r(\hat{\pi}) - J_r(\mu)) = \mathcal{L}_\mu(\hat{\pi}, Q_r^{\hat{\pi}})$$
$$= \Delta(\hat{\pi}) + \mathcal{L}_\mu(\hat{\pi}, f) \qquad (\Delta(\hat{\pi}) := \mathcal{L}_\mu(\hat{\pi}, Q_r^{\hat{\pi}}) - \mathcal{L}_\mu(\hat{\pi}, f))$$
$$= \Delta(\hat{\pi}) + \mathbb{E}_\mu[f(s,\hat{\pi}) - f(s,a)]$$
$$= \Delta(\hat{\pi}) + (1-\gamma)(J_{R^{f,\hat{\pi}}}(\hat{\pi}) - J_{R^{f,\hat{\pi}}}(\mu))$$
$$= \Delta(\hat{\pi}) + (1-\gamma)Q_{R^{f,\hat{\pi}}}^{\hat{\pi}}(s_0, \hat{\pi}) - \mathbb{E}_\mu[R^{\hat{\pi},f}(s,a)]$$
$$= \Delta(\hat{\pi}) + (1-\gamma)f(s_0, \hat{\pi}) - \mathbb{E}_\mu[R^{\hat{\pi},f}(s,a)],$$

where the last equality is true because that

$$Q_{R^{f,\hat{\pi}}}^\pi(s,a) = (\mathcal{T}_{R^{f,\hat{\pi}}}^\pi f)(s,a) = R^{f,\hat{\pi}} + \gamma \mathbb{E}_{s'|(s,a)}[f(s',\hat{\pi})] = f(s,a).$$

Thus we have

$$(J_r(\pi) - J_r(\hat{\pi})) = (J_r(\pi) - J_r(\mu) - (J_r(\hat{\pi}) - J_r(\mu)))$$
$$= (J_r(\pi) - f(d_0, \hat{\pi})) + \left(\mathbb{E}_\mu[R^{\hat{\pi},f}(s,a)] - J_r(\mu)\right) - \Delta(\hat{\pi}). \tag{25}$$

For the first term, we have

$$
\begin{aligned}
(J_r(\pi) - f(d_0, \hat{\pi})) =& (J_r(\pi) - f(s_0, \hat{\pi})) && \text{(deterministic initial state)} \\
=& J_r(\pi) - \mathbb{E}_{d^\pi}[R^{\hat{\pi},f}(s,a)] + \mathbb{E}_{d^\pi}[R^{\hat{\pi},f}(s,a)] - f(s_0, \hat{\pi}) \\
=& \mathbb{E}_{d^\pi}[R(s,a) - R^{\hat{\pi},f}(s,a)] + \mathbb{E}_{d^\pi}[f(s,\pi) - f(s,\hat{\pi})] \\
=& \mathbb{E}_{d^\pi}[(\mathcal{T}_r^{\hat{\pi}} f - f)(s,a)] + \mathbb{E}_{d^\pi}[f(s,\pi) - f(s,\hat{\pi})], && (26)
\end{aligned}
$$

where the second equality is true because

$$
\begin{aligned}
& \mathbb{E}_{d^\pi}[R^{\hat{\pi},f}(s,a)] - f(s_0, \hat{\pi}) \\
=& \mathbb{E}_{d^\pi}\left[f(s,a) - \gamma \mathbb{E}_{s'|(s,a)}[f(s', \hat{\pi})]\right] - f(s_0, \hat{\pi}) \\
=& \mathbb{E}_{d^\pi}\left[f(s,\pi)\right] - \sum_s \sum_{t=1}^{\infty} \gamma^t \Pr(s_t = s | s_0 \sim d_0, \pi) f(s, \hat{\pi}(s)) - f(s_0, \hat{\pi}) \\
=& \mathbb{E}_{d^\pi}\left[f(s,\pi)\right] - \sum_s \sum_{t=0}^{\infty} \gamma^t \Pr(s_t = s | s_0 \sim d_0, \pi) f(s, \hat{\pi}(s)) \\
=& \mathbb{E}_{d^\pi}\left[f(s,\pi)\right] - \sum_{s,a} \sum_{t=0}^{\infty} \gamma^t \Pr(s_t = s, a_t = a | s_0 \sim d_0, \pi) f(s, \hat{\pi}(s)) \\
=& \mathbb{E}_{d^\pi}[f(s,\pi) - f(s,\hat{\pi})].
\end{aligned}
$$

For the second term we have

$$
\begin{aligned}
& \mathbb{E}_\mu[R^{\hat{\pi},f}(s,a)] - J_r(\mu) \\
=& \mathbb{E}_\mu[R^{\hat{\pi},f}(s,a) - R(s,a)] \\
=& \mathbb{E}_\mu[(f - \mathcal{T}_r^{\hat{\pi}} f)(s,a)]. && (27)
\end{aligned}
$$

Therefore plugging 26 and (27) into Eq. (25), we have

$$
\begin{aligned}
& J_r(\pi) - J_r(\hat{\pi}) \\
=& \mathbb{E}_\mu\left[(f - \mathcal{T}_r^{\hat{\pi}})(s,a)\right] + \mathbb{E}_\pi\left[(\mathcal{T}_r^{\hat{\pi}} f - f)(s,a)\right] + \mathbb{E}_\pi[f(s,\pi) - f(s,\hat{\pi})] + \mathcal{L}_\mu(\hat{\pi}, f) - \mathcal{L}_\mu(\hat{\pi}, Q_r^{\hat{\pi}}).
\end{aligned}
$$

The proof is completed. $\square$

**Lemma 7.** *With probability at least $1 - 2\delta$, for any $f_r \in \mathcal{F}, f_c \in \mathcal{G}$, and $\pi \in \Pi$, we have:*

$$
|\mathcal{L}_\mu(\pi, f_r) - \mathcal{L}_\mathcal{D}(\pi, f_r)| \le \epsilon_{stat} \tag{28}
$$

$$
|\mathcal{L}_\mu(\pi, f_c) - \mathcal{L}_\mathcal{D}(\pi, f_c)| \le \epsilon_{stat} \tag{29}
$$

*where $\epsilon_{stat} := V_{\max} C_{\ell_2}^* \sqrt{\frac{\log(|\mathcal{F}||\mathcal{G}||\Pi||W|/\delta)}{N}} + \frac{V_{\max} B_w \log(|\mathcal{F}||\mathcal{G}||\Pi||W|/\delta)}{N}$.*

*Proof.* Recall that $\mathbb{E}_\mu[\mathcal{L}_\mathcal{D}(\pi, f_r)] = \mathcal{L}_\mu(\pi, f)$ and $|f_r(s,\pi) - f_r(s,a)| \le V_{\max}$. For any $f_r \in \mathcal{F}$, policy $\pi \in \Pi$, applying a Hoeffding's inequality and a union bound we can obtain with probability $1 - \delta$,

$$
|\mathcal{L}_\mu(\pi, f_r) - \mathcal{L}_\mathcal{D}(\pi, f_r)| \le \mathcal{O}\left(V_{\max} \sqrt{\frac{\log(|\mathcal{F}||\Pi|/\delta)}{N}}\right) \le \epsilon_{stat}. \tag{30}
$$

The inequality for proving the $f_c, \pi$ is the same. $\square$

# B  Missing Proofs

## B.1  Proof of Theorem 4.1

*Proof.* According to the performance difference lemma (Kakade and Langford, 2002), we have

$$
\begin{aligned}
& (J_r(\pi) - J_r(\mu)) - \lambda\{J_c(\pi) - J_c(\mu)\}_+ \\
=& \mathcal{L}_\mu(\pi, Q_r^\pi) - \lambda\{\mathcal{L}_\mu(\pi, Q_c^\pi)\}_+ \\
=& \mathcal{L}_\mu(\pi, Q_r^\pi) + \beta\mathcal{E}_\mu(\pi, Q_r^\pi) - \lambda\{\mathcal{L}_\mu(\pi, Q_c^\pi)\}_+ + \beta\hat{\mathcal{E}}_\mu(\pi, Q_c^\pi) \\
\ge& \mathcal{L}_\mu(\pi, f_r^\pi) + \beta\mathcal{E}_\mu(\pi, f_r^\pi) - \lambda\{\mathcal{L}_\mu(\pi, f_c^\pi)\}_+ + \beta\hat{\mathcal{E}}_\mu(\pi, f_c^\pi) \\
\ge& \mathcal{L}_\mu(\pi, f_r^\pi) - \lambda\{\mathcal{L}_\mu(\pi, f_c^\pi)\}_+, && (31)
\end{aligned}
$$

where the second equality is true because $\mathcal{E}_\mu(\pi, Q_r^\pi) = \hat{\mathcal{E}}_\mu(\pi, Q_c^\pi) = 0$ by Assumption 3.2, and the first inequality comes from the selection of $f_r^\pi$ and $f_c^\pi$ in optimization (6).

Therefore, we can obtain

$$
\begin{aligned}
J_r(\hat{\pi}^*) - J_r(\mu) &\geq \left(\mathcal{L}_\mu(\hat{\pi}^*, f_r^{\hat{\pi}^*}) - \lambda\{\mathcal{L}_\mu(\hat{\pi}^*, f_c^{\hat{\pi}^*})\}_+\right) + \lambda\{J_c(\hat{\pi}^*) - J_c(\mu)\}_+ \\
&\geq \left(\mathcal{L}_\mu(\mu, f_r^\mu) - \lambda\{\mathcal{L}_\mu(\mu, f_c^\mu)\}_+\right) + \lambda\{J_c(\hat{\pi}^*) - J_c(\mu)\}_+ \\
&\geq \lambda\{J_c(\hat{\pi}^*) - J_c(\mu)\}_+ \geq 0
\end{aligned}
\tag{32}
$$

and

$$
\{J_c(\hat{\pi}^*)\}_+ - \{J_c(\mu)\}_+ \leq \{J_c(\hat{\pi}^*) - J_c(\mu)\}_+ \leq \frac{1}{\lambda}(J_r(\hat{\pi}^*) - J_r(\mu)) \leq \frac{1}{\lambda}.
\tag{33}
$$

$\square$

## B.2  Proof of Lemma 2

*Proof.* Denote $\pi_{\text{ref}}$ as $\pi$. First according to the definition for the no-regret oracle 5.1, we have

$$
\begin{aligned}
\frac{1}{K}\sum_{k=1}^K \mathbb{E}_\pi[f_r^k(s,\pi) - f_r^k(s,\pi_k) &- \lambda\{f_c^k(s,\pi) - f_c^k(s,\pi)\}_+ \\
&+ \lambda\{f_c^k(s,\pi_k) - f_c^k(s,\pi)\}_+] \leq \epsilon_{opt}^\pi
\end{aligned}
\tag{34}
$$

Therefore,

$$
\frac{1}{K}\sum_{k=1}^K \mathbb{E}_\pi[f_r^k(s,\pi) - f_r^k(s,\pi_k)]
$$

$$
\leq \epsilon_{opt}^\pi + \frac{1}{K}\sum_{k=1}^K \mathbb{E}_\pi[\lambda\{f_c^k(s,\pi) - f_c^k(s,\pi)\}_+ - \lambda\{f_c^k(s,\pi_k) - f_c^k(s,\pi)\}_+] \leq \epsilon_{opt}^\pi,
\tag{35}
$$

and

$$
\frac{1}{K}\sum_{k=1}^K \mathbb{E}_\pi[\{f_c^k(s,\pi_k) - f_c^k(s,\pi)\}_+] \leq \epsilon_{opt}^\pi - \frac{1}{\lambda K}\sum_{k=1}^K \mathbb{E}_\pi[f_r^k(s,\pi) - f_r^k(s,\pi_k)] \leq \epsilon_{opt}^\pi + \frac{V_{\max}}{\lambda}.
\tag{36}
$$

We finish the proof. $\square$

## B.3  Proof of Theorem 5.2

**Theorem** (Restate of Theorem 5.2). *Under Assumptions 3.2 and 3.6, let the reference policy $\pi_{ref} \in \Pi$ be any policy satisfying Assumption 3.7, then with probability at least $1 - \delta$,*

$$
J_r(\pi_{ref}) - J_r(\bar{\pi}) \leq \mathcal{O}\left(\epsilon_{stat} + C_{\ell_2}^* \sqrt{\epsilon_1}\right) + \epsilon_{opt}^\pi
\tag{37}
$$

$$
J_c(\bar{\pi}) - J_c(\pi_{ref}) \leq \mathcal{O}\left(\epsilon_{stat} + C_{\ell_2}^* \sqrt{\epsilon_1}\right) + \epsilon_{opt}^\pi + \frac{V_{\max}}{\lambda},
\tag{38}
$$

*where $\epsilon_{stat} := V_{\max} C_{\ell_2}^* \sqrt{\frac{\log(|\mathcal{F}||\mathcal{G}||\Pi||W|/\delta)}{N}} + \frac{V_{\max} B_w \log(|\mathcal{F}||\mathcal{G}||\Pi||W|/\delta)}{N}$, and $\bar{\pi}$ is the policy returned by Algorithm 1 with $\beta > 0$ and $\pi_{ref}$ as input.*

*Proof.* Denote $\pi_{\text{ref}}$ as $\pi$. According to the definition of $\bar{\pi}$, and Lemma 6 we have

$$J_r(\pi) - J_r(\bar{\pi}) = \frac{1}{K} \sum_{k=1}^{K} (J_r(\pi) - J_r(\pi_k))$$

$$= \frac{1}{K} \sum_{k=1}^{K} \left( \underbrace{\mathbb{E}_\mu \left[ f_r^k - \mathcal{T}_r^{\pi_k} f_r^k \right]}_{\text{(I)}} + \underbrace{\mathbb{E}_\pi [\mathcal{T}_r^{\pi_k} f_r^k - f_r^k]}_{\text{(II)}} \right.$$

$$\left. + \underbrace{\mathbb{E}_\pi [f_r^k(s,\pi) - f_r^k(s,\pi_k)]}_{\text{(III)}} + \underbrace{\mathcal{L}_\mu(\pi_k, f_r^k) - \mathcal{L}_\mu(\pi_k, Q^{\pi_k})}_{\text{(IV)}} \right) \tag{39}$$

Condition on the high probability event in , we have

$$\text{(I)} + \text{(II)} \le 2\mathcal{E}_\mu(\pi_k, f_r^k) \le 2\mathcal{E}_\mathcal{D}(\pi_k, f_r^k) + 2\epsilon_{stat} \tag{40}$$

According to a similar argument as that in the Lemma 13 in Cheng et al. (2022), we have that

$$|\mathcal{L}_\mu(\pi_k, Q_r^{\pi_k}) - \mathcal{L}_\mu(\pi_k, f_r^{\pi_k})|$$
$$= |\mathbb{E}_\mu[Q_r^{\pi_k}(s, \pi_k) - Q_r^{\pi_k}(s, a)] - \mathcal{L}_\mu(\pi_k, f_r^{\pi_k})|$$
$$= |(J_r(\pi_k) - J_r(\mu)) - \mathcal{L}_\mu(\pi_k, f_r^{\pi_k})|$$
$$= |(f_r^{\pi_k}(s_0, \pi_k) - J_r(\mu)) + (J_r(\pi_k) - f_r^{\pi_k}(s_0, \pi_k)) - \mathcal{L}_\mu(\pi_k, f_r^{\pi_k})|$$
$$= |\mathbb{E}_\mu[f_r^{\pi_k}(s, \pi_k) - (\mathcal{T}_r^{\pi_k} f_r^{\pi_k})(s, a)] + \mathbb{E}_{d^{\pi_k}}[(\mathcal{T}_r^{\pi_k} f_r^{\pi_k})(s, a) - f_r^{\pi_k}(s, a)] - \mathcal{L}_\mu(\pi_k, f_r^{\pi_k})|$$
$$\text{(by the extension of performance difference lemma (Lemma 1 in Cheng et al. (2020)))}$$
$$= |\mathcal{L}_\mu(\pi_k, f_r^{\pi_k}) + \mathbb{E}_\mu[f_r^{\pi_k}(s, a) - (\mathcal{T}_r^{\pi_k} f_r^{\pi_k})(s, a)] + \mathbb{E}_{d^{\pi_k}}[(\mathcal{T}_r^{\pi_k} f_r^{\pi_k})(s, a) - f_r^{\pi_k}(s, a)] - \mathcal{L}_\mu(\pi_k, f_r^{\pi_k})|$$
$$\le \|f_r^{\pi_k}(s, a) - (\mathcal{T}_r^{\pi_k} f_r^{\pi_k})(s, a)\|_{2,\mu} + \|(\mathcal{T}_r^{\pi_k} f_r^{\pi_k})(s, a) - f_r^{\pi_k}(s, a)\|_{2,d^{\pi_k}}$$
$$\le \mathcal{O}(\sqrt{\epsilon_1}), \tag{41}$$

where $f_r^\pi := \arg\min_{f_r \in \mathcal{F}} \sup_{\text{admissible } \nu} \|f_r - \mathcal{T}_r^\pi f_r\|_{2,\nu}^2, \forall \pi \in \Pi$. By using Lemma 7, we have

$$|\mathcal{L}_\mu(\pi_k, f_r^k) - \mathcal{L}_\mathcal{D}(\pi_k, f_r^k)| + |\mathcal{L}_\mu(\pi_k, f_r^{\pi_k}) - \mathcal{L}_\mathcal{D}(\pi_k, f_r^{\pi_k})| \le \mathcal{O}(\epsilon_{stat}). \tag{42}$$

Therefore

$$\text{(I)} + \text{(II)} + \text{(IV)} \le \mathcal{L}_\mu(\pi_k, f_r^k) + 2\mathcal{E}_\mu(\pi_k, f_r^k) + 2\epsilon_{stat} - \mathcal{L}_\mu(\pi_k, f_r^{\pi_k}) + \mathcal{O}(\sqrt{\epsilon_1}) \tag{43}$$

$$\le \mathcal{L}_\mathcal{D}(\pi_k, f_r^k) + 2\mathcal{E}_\mathcal{D}(\pi_k, f_r^k) + \mathcal{O}(\epsilon_{stat}) - \mathcal{L}_\mathcal{D}(\pi_k, f_r^{\pi_k}) + \mathcal{O}(\sqrt{\epsilon_1}) \tag{44}$$

$$\le \mathcal{L}_\mathcal{D}(\pi_k, f_r^{\pi_k}) + 2\mathcal{E}_\mathcal{D}(\pi_k, f_r^{\pi_k}) + \mathcal{O}(\epsilon_{stat}) - \mathcal{L}_\mathcal{D}(\pi_k, f_r^{\pi_k}) + \mathcal{O}(\sqrt{\epsilon_1}) \tag{45}$$

$$\le \mathcal{O}(\epsilon_{stat} + C_{\ell_2}^* \sqrt{\epsilon_1}), \tag{46}$$

where the third inequality holds by the selection of $f_r^k$, and the last inequality holds by Lemma 5. Therefore by using Lemma B.2 we obtain

$$J_r(\pi) - J_r(\bar{\pi}) \le \mathcal{O}(\epsilon_{stat} + C_{\ell_2}^* \sqrt{\epsilon_1}) + \epsilon_{opt}. \tag{47}$$

Following a similar argument, we have that

$$J_c(\bar{\pi}) - J_c(\pi) = \frac{1}{K} \sum_{k=1}^{K} (J_c(\pi_k) - J_c(\pi)) \le \mathcal{O}(\epsilon_{stat} + C_{\ell_2}^* \sqrt{\epsilon_1}) + \epsilon_{opt}^\pi + \frac{V_{max}}{\lambda}. \tag{48}$$

$$\square$$

## B.4  Proof of Theorem 5.6

**Theorem** (Restate of Theorem 5.6). *Under Assumptions 3.2 and 3.6, let the reference policy $\pi_{ref} \in \Pi$ be any policy satisfying Assumption 3.7, then with probability at least $1 - \delta$,*

$$J_r(\mu) - J_r(\bar{\pi}) \le \mathcal{O}\left( \epsilon_{stat}^\pi + C_{\ell_2}^* \sqrt{\epsilon_1} \right) + \epsilon_{opt}^\mu \tag{49}$$

$$J_c(\bar{\pi}) - J_c(\mu) \le \mathcal{O}\left( \epsilon_{stat}^\pi + C_{\ell_2}^* \sqrt{\epsilon_1} \right) + \epsilon_{opt}^\mu + \frac{V_{\max}}{\lambda}, \tag{50}$$

where $\epsilon_{stat} := V_{\max} C^*_{\ell_2} \sqrt{\frac{\log(|\mathcal{F}||\Pi||W|/\delta)}{N}} + \frac{V_{\max} B_w \log(|\mathcal{F}||\Pi||W|/\delta)}{N}$, and $\bar{\pi}$ is the policy returned by Algorithm 1 with $\beta \geq 0$ and $\mu$ as input.

*Proof.* Following a similar proof in Theorem 5.2. But when the reference policy is the behavior policy, we have $(I) + (II) = 0$. Therefore we have have

$$(IV) = \mathcal{L}_\mu(\pi_k, f_r^k) - \mathcal{L}_\mu(\pi_k, Q^{\pi_k})$$

$$\leq \mathcal{L}_\mu(\pi_k, f_r^k) - \mathcal{L}_\mu(\pi_k, Q^{\pi_k}) + \beta \mathcal{E}_\mathcal{D}(\pi_k, f_r^k)$$

$$\leq \mathcal{L}_\mu(\pi_k, f_r^k) - \mathcal{L}_\mu(\pi_k, Q^{\pi_k}) + \beta \mathcal{E}_\mathcal{D}(\pi_k, f_r^k) - \beta \mathcal{E}_\mathcal{D}(\pi, f_{\pi_k}) + \beta C^*_{\ell_2} \sqrt{\epsilon_1} + \beta \epsilon_{stat} \quad \text{(Lemma 5)}$$

$$\leq \mathcal{L}_\mathcal{D}(\pi_k, f_r^k) + \beta \mathcal{E}_\mathcal{D}(\pi_k, f_r^k) - \mathcal{L}_\mathcal{D}(\pi_k, f_r^{\pi_k}) - \beta \mathcal{E}_\mathcal{D}(\pi, f_{\pi_k}) + (\beta + 1)(\epsilon_{stat} + C^*_{\ell_2} \sqrt{\epsilon_1})$$

$$\leq (\beta + 1)(\epsilon_{stat} + C^*_{\ell_2} \sqrt{\epsilon_1}).$$

We finish the proof. $\qquad \square$

## C  Discussion on obtaining the behavior policy

To extract the behavior policy when it is not provided, we can simply run behavior cloning on the offline data. In particular, we can estimate the learned behavior policy $\hat{\pi}_\mu$ as follows: $\forall s \in \mathcal{D}, \hat{\pi}_\mu(a|s) \leftarrow \frac{n(s,a)}{n(s)}$, and $\forall s \notin \mathcal{D}, \hat{\pi}_\mu(a|s) \leftarrow \frac{1}{|\mathcal{A}|}$, where $n(s,a)$ is the number of times $(s,a)$ appears in the offline dataset $\mathcal{D}$. Essentially, the estimated BC policy matches the empirical behavior policy on states in the offline dataset and takes uniform random actions outside the support of the dataset. It is easy to show that the gap between the learned policy $\hat{\pi}_\mu$ and the behavior policy $\pi_\mu$ is upper bounded by $\mathcal{O}(\min\{1, |\mathcal{S}|/N\})$ (Kumar et al., 2022; Rajaraman et al., 2020). We can have a very accurate estimate as long as the size of the dataset is large enough.

## D  Experimintal Supplement

### D.1  Practical Algorithm

The practical version of our algorithm WSAC is shown in Algorithm 2.

---
**Algorithm 2** WSAC - Practical Version
---
1: **Input:** Batch data $\mathcal{D}$, policy network $\pi$, network for the reward critic $f_r$, network for the cost critic $f_c, \beta > 0, \lambda > 0$.
2: **for** $k = 1, 2, \ldots, K$ **do**
3:     Sample minibatch $\mathcal{D}_{\text{mini}}$ from the dataset $\mathcal{D}$.
4:     Update Critic Networks:

$$l_{\text{reward}}(f_r) := \mathcal{L}_{\mathcal{D}_{\text{mini}}}(\pi, f_r) + \beta \mathcal{E}_{\mathcal{D}_{\text{mini}}}(\pi, f_r),$$
$$f_r \leftarrow \text{ADAM}(f_r - \eta_{\text{fast}} \nabla l_{\text{reward}}(f_r)),$$
$$l_{\text{cost}}(f_c) := -\lambda \mathcal{L}_{\mathcal{D}_{\text{mini}}}(\pi, f_c) + \beta \mathcal{E}_{\mathcal{D}_{\text{mini}}}(\pi, f_c),$$
$$f_c \leftarrow \text{ADAM}(f_c - \eta_{\text{fast}} \nabla l_{\text{cost}}(f_c)).$$

5:     Update Policy Network:

$$l_{\text{actor}}(\pi) := -\mathcal{L}_{\text{mini}}(\pi, f_r) + \lambda \{\mathcal{L}_{\text{mini}}(\pi, f_c)\}_+,$$
$$\pi \leftarrow \text{ADAM}(\pi - \eta_{\text{slow}} \nabla l_{\text{actor}}(\pi)).$$

6: **end for**
7: **Output:** $\pi$

---

### D.2  Environments Description

Besides the "BallCircle" environment, we also study several representative environments as follows. All of them are shown in Figure 2 and their offline dataset is from Liu et al. (2023a).

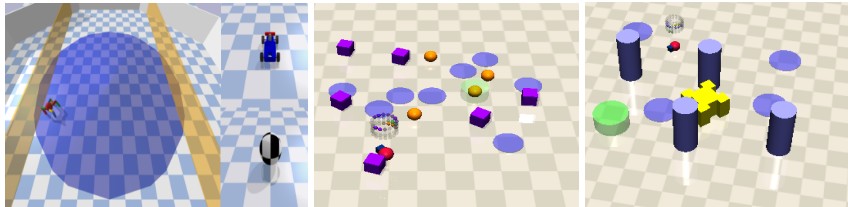

Figure 2: BallCircle and CarCircle (left), PointButton (medium), PointPush(right) .

- **CarCircle**: This environment requires the car to move on a circle in a clockwise direction within the safety zone defined by the boundaries. The car is a four-wheeled agent based on MIT's race car. The reward is dense and increases by the car's velocity and by the proximity towards the boundary of the circle and the cost is incurred if the agent leaves the safety zone defined by the two yellow boundaries, which are the same as "CarCircle".

- **PointButton**: This environment requires the point to navigate to the goal button location and touch the right goal button while avoiding more gremlins and hazards. The point has two actuators, one for rotation and the other for forward/backward movement. The reward consists of two parts, indicating the distance between the agent and the goal and if the agent reaches the goal button and touches it. The cost will be incurred if the agent enters the hazardous areas, contacts the gremlins, or presses the wrong button.

- **PointPush**: This environment requires the point to push a box to reach the goal while circumventing hazards and pillars. The objects are in 2D planes and the point is the same as "PointButton". It has a small square in front of it, which makes it easier to determine the orientation visually and also helps point push the box.

### D.3 Implementation Details and Experimental settings

We run all the experiments with NVIDIA GeForce RTX $3080$ Ti $8-$Core Processor.

The normalized reward and cost are summarized as follows:

$$R_{normalized} = \frac{R_\pi - r_{min}(\mathcal{M})}{r_{max}(\mathcal{M}) - r_{min}(\mathcal{M})} \quad (51)$$

$$C_{normalized} = \frac{C_\pi + \epsilon}{\kappa + \epsilon}, \quad (52)$$

where $r(\mathcal{M})$ is the empirical reward for task $\mathcal{M}$, $\kappa$ is the cost threshold, $\epsilon$ is a small number to ensure numerical stability. Thus any normalized cost below $1$ is considered as safe. We use $R_\pi$ and $C_\pi$ to dentoe the cumulative rewards and cost for the evaluated policy, respectively. The parameters of $r_{min}(\mathcal{M})$, $r_{max}(\mathcal{M})$ and $\kappa$ are environment-dependent constants and the specific values can be found in the Appendix D. We remark that the normalized reward and cost only used for demonstrating the performance purpose and are not used in the training process. The detailed value of the reward and costs can be found in Table 3. To mitigate the risk of unsafe scenarios, we introduce a hyperparameter

Table 3: Hyperparameters of WSAC

| Parameters | BallCircle | CarCircle | PointButton | PointPush |
|---|---|---|---|---|
| $\beta_c$ | 30.0 | 38.0 | 30.0 | 30.0 |
| $\beta_r$ | 10.0 | 12.0 | 10.0 | 10.0 |
| $UB_{Q_C}$ | 30.0 | 28.0 | 32.0 | 30.0 |
| $\lambda$ | [1.0, 20.0] | | | |
| Batch size | 512 | | | |
| Actor learning rate | 0.0001 | | | |
| Critic learning rate | 0.0003 | | | |
| $\kappa$ | 40 | | | |
| $r_{min}(\mathcal{M})$ | 0.3831 | 3.4844 | 0.0141 | 0.0012 |
| $r_{max}(\mathcal{M})$ | 881.4633 | 534.3061 | 42.8986 | 14.6910 |

$UB_{Q_C}$ to the cost $Q$-function as an overestimation when calculating the actor loss. We use two separate $\beta_r$, $\beta_c$ for reward and cost $Q$ functions to make the algorithm more flexible.

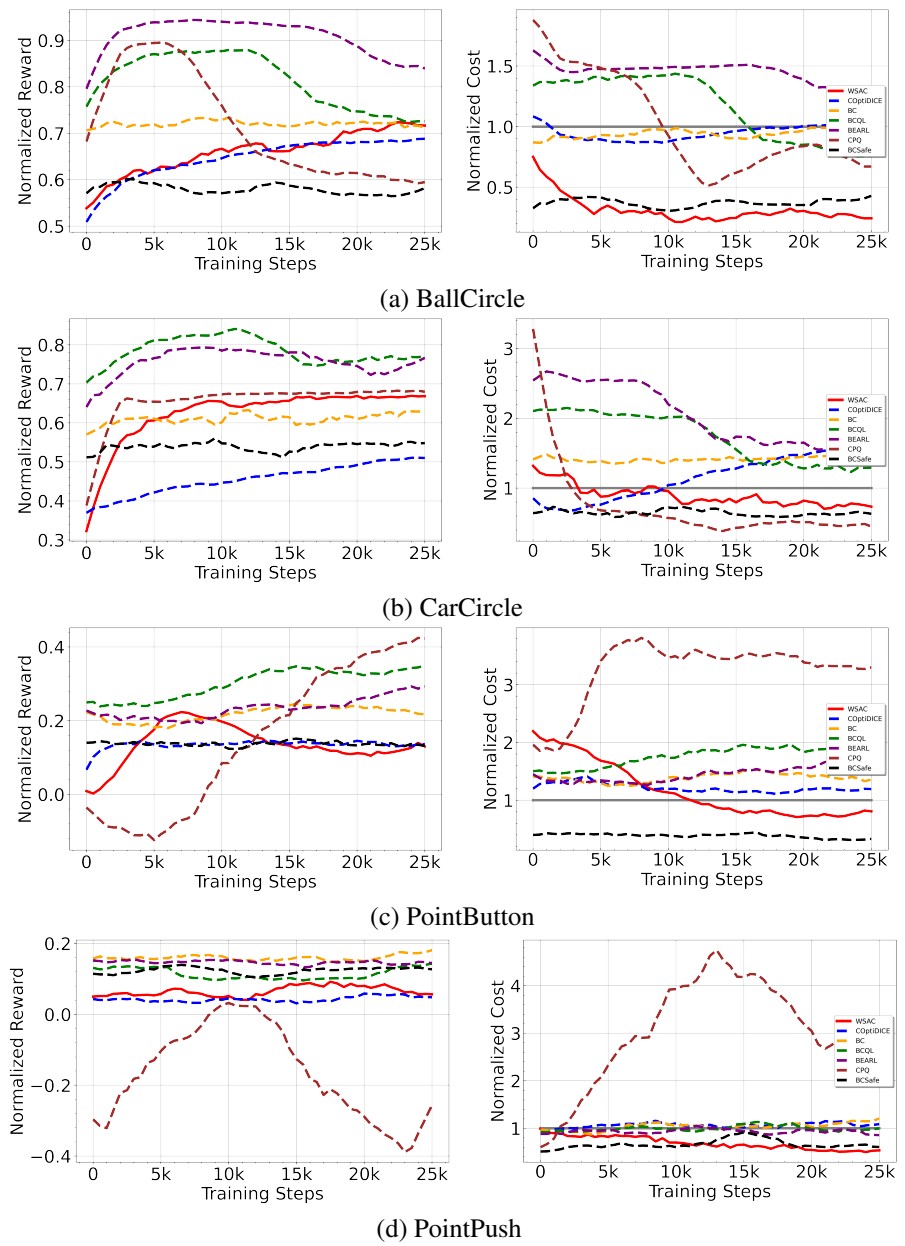

Figure 3: The moving average of evaluation results is recorded every 500 training steps, with each result representing the average over 20 evaluation episodes and three random seeds. A cost threshold 1 is applied, with any normalized cost below 1 considered safe.

We use different $\beta$ for the reward and cost critic networks and different $UB_{Q_C}$ for the actor-network to make the adversarial training more stable. We also let the key parameter $\lambda$ within a certain range balance reward and cost during the training process. Their values are shown in Table 3. In experiments, we take $\mathcal{W} = \{0, C_\infty\}$ for computation effective. Then we can reduce $\mathcal{E}_\mathcal{D}(\pi, f)$ to $C_\infty \mathbb{E}_\mathcal{D}[(f(s,a) - r - \gamma f(s', \pi))^2]$ and reduce $\hat{\mathcal{E}}_\mathcal{D}(\pi, f)$ to $C_\infty \mathbb{E}_\mathcal{D}[(f(s,a) - c - \gamma f(s', \pi))^2]$. In this case, $C_\infty$ can be considered as a part of the hyperparameter $\beta_r (\beta_c)$.

### D.4 Experimental results details and supplements

The evaluation performances of the agents in each environment after 30000 update steps of training are shown in Table 2, and the performance of average rewards and costs are shown in Figure 3. From the results, we observe that WSAC achieves a best reward performance with significantly lowest costs against all the baselines. It suggests WSAC can establish a safe and efficient policy and achieve a steady improvement by leveraging the offline dataset.

### D.5 Simulations under different cost limits

To further evaluate the performance of our algorithm under varying situations. We further compare our algorithm with baselines under varying cost limits, we report the average performance of our method and other baselines in Table 4. Specifically, cost limits of $[10, 20, 40]$ are used for the BallCircle and CarCircle environments, and $[20, 40, 80]$ for the PointButton and PointPush environments, following the standard setup outlined by Liu et al. (2023a). Our results demonstrate that WSAC maintains safety across all environments, and its performance is either comparable to or superior to the best baseline in each case. These suggest that WSAC is well-suited for adapting to tasks of varying difficulty.

Table 4: The normalized reward and cost of WSAC and other baselines for different cost limits. Each value is averaged over 3 distinct cost limits, 20 evaluation episodes, and 3 random seeds. The Average line shows the average situation in various environments. The cost threshold is 1. Gray: Unsafe agent whose normalized cost is greater than 1. Blue: Safe agent with best performance. The performance of all the baselines is reported according to the results in Liu et al. (2023a).

|  | BC | | Safe-BC | | CDT | | BCQL | | BEARL | | CPQ | | COptiDICE | | WSAC | |
|---|---|---|---|---|---|---|---|---|---|---|---|---|---|---|---|---|
|  | Reward ↑ | Cost ↓ | Reward ↑ | Cost ↓ | Reward ↑ | Cost ↓ | Reward ↑ | Cost ↓ | Reward ↑ | Cost ↓ | Reward ↑ | Cost ↓ | Reward ↑ | Cost ↓ | Reward ↑ | Cost ↓ |
| BallCircle | 0.74 | 4.71 | 0.52 | 0.65 | 0.77 | 1.07 | 0.69 | 2.36 | 0.86 | 3.09 | 0.64 | 0.76 | 0.70 | 2.61 | 0.74 | 0.51 |
| CarCircle | 0.58 | 3.74 | 0.50 | 0.84 | 0.75 | 0.95 | 0.63 | 1.89 | 0.74 | 2.18 | 0.71 | 0.33 | 0.49 | 3.14 | 0.65 | 0.55 |
| PointButton | 0.27 | 2.02 | 0.16 | 1.10 | 0.46 | 1.57 | 0.40 | 2.66 | 0.43 | 2.47 | 0.58 | 4.30 | 0.15 | 1.51 | 0.11 | 0.55 |
| PointPush | 0.18 | 0.91 | 0.11 | 0.80 | 0.21 | 0.65 | 0.23 | 0.99 | 0.16 | 0.89 | 0.11 | 1.04 | 0.02 | 1.18 | 0.07 | 0.61 |
| Average | 0.44 | 2.85 | 0.32 | 0.85 | 0.55 | 1.06 | 0.49 | 1.98 | 0.55 | 2.16 | 0.51 | 1.61 | 0.34 | 2.11 | 0.39 | 0.56 |

### D.6 Ablation studies

To investigate the contribution of each component of our algorithm, including the weighted Bellman regularizer, the aggression-limited objective, and the no-regret policy optimization (which together guarantee our theoretical results), we performed an ablation study in the tabular setting. The results, presented in Table 5, indicate that the weighted Bellman regularization ensures the safety of the algorithm, while the aggression-limited objective fine-tunes the algorithm to achieve higher rewards without compromising safety.

Table 5: Ablation study under tabular case (cost limit is 0.1) over 10 repeat experiments

| Components | cost | reward |
|---|---|---|
| ALL | $0.014 \pm 0.006$ | $0.788 \pm 0.004$ |
| W/O no-regret policy optimization | $0.014 \pm 0.006$ | $0.788 \pm 0.004$ |
| W/O Aggression-limited objective | $0.014 \pm 0.006$ | $0.788 \pm 0.005$ |
| W/O Weighted Bellman regularizer | $0.323 \pm 0.061$ | $0.684 \pm 0.017$ |

### D.7 Sensitivity Analysis of Hyper-Parameters

We provide the rewards and costs under different sets of $\beta_r = \beta_c \in \{1, 0.5, 0.05\}$ and $\lambda \in \{[0,1], [0,2], [1,2]\}$ (since our $\lambda$ only increases, the closed interval here represents the initial value

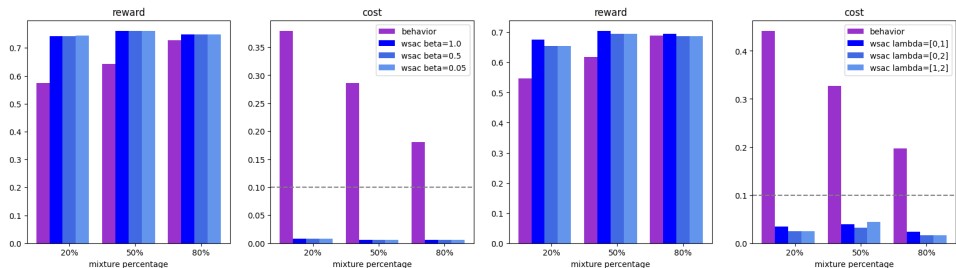

Figure 4: Sensitivity Analysis of Hyperparameters in the Tabular Case. The left figure illustrates tests conducted with various $\beta$ values (For the sake of discussion, we denote $\beta = \beta_r = \beta_c$) with $\lambda = [0, 2]$, while the right figure presents tests across different ranges of $\lambda$ with $\beta_r = \beta_c = 2.0$. and the upper bound of $\lambda$) to demonstrate the robustness of our approach in the tabular setting in Figure 4. We can observe that the performance is almost the same under different sets of parameters and different qualities of behavior policies.

