# OpenReview forum: "Adversarially Trained Weighted Actor-Critic for Safe Offline Reinforcement Learning"
_NeurIPS.cc/2024/Conference — NeurIPS 2024 poster_

### Official Review · Reviewer_HDvF · 2024-06-30

**Soundness:** 2
**Presentation:** 2
**Contribution:** 2
**Rating:** 5
**Confidence:** 2

**Summary:**

The Weighted Safe Actor-Critic (WSAC) is a new Safe Offline Reinforcement Learning algorithm designed to outperform any reference policy while ensuring safety with limited data. It uses a two-player Stackelberg game to achieve optimal convergence and safe policy improvements. In practical tests, WSAC surpasses baselines in continuous control environments.

**Strengths:**

- The ability of WSAC to outperform the behavior policy over a wide range of hyperparameters is a crucial property for practical use.
- The author provides theoretical proof.

**Weaknesses:**

- Given that I haven't examined the mathematical details, I find that many of the assumptions and proofs of key theorems in the paper resemble those in ATAC [1]. The primary differences are the authors' focus on the safe offline RL setting and the inclusion of a cost value in their theory. However, the use of a primal-dual approach in the algorithm's implementation may introduce training stability issues [2]. From both theoretical and practical implementation perspectives, it is difficult to identify novel insights in the paper.
- The author needs to compare more state-of-the-art baselines, such as CDT [3] and FISOR [2].
- Line 34 contains a duplicate citation.

[1] Cheng, Ching-An, et al. "Adversarially trained actor critic for offline reinforcement learning." *International Conference on Machine Learning*. PMLR, 2022.

[2] Zheng, Yinan, et al. "Safe offline reinforcement learning with feasibility-guided diffusion model". *International Conference on Learning Representations* (2024).

[3] Liu, Zuxin, et al. "Constrained decision transformer for offline safe reinforcement learning." *International Conference on Machine Learning*. PMLR, 2023.

**Questions:**

- What about the performance under different cost limits? The average cost in Table 2 does not adequately reveal the safety of the algorithm.

---

> ### Author Rebuttal · Authors · 2024-08-07
>
> Thank you for your comments on our paper. Please find our point-by-point response to your questions below.
>
> - **Response to contributions:** We respectfully ask the reviewer to evaluate our theoretical contributions. We would like to mention that our approach has significant differences from ATAC. It is also unfair to say the "primary differences are the authors' focus on the safe offline RL setting and the inclusion of a cost value in their theory," since all existing studies in safe RL consider the cost function in the formulation of RL, it is the default setting. This consideration makes the problem entirely different and more difficult than the unconstrained case; i.e., the optimal policy will no longer be a greedy policy, how to ensure safe learning and obtain a safe policy, and balancing the reward and cost is extremely important. We are the **first** to present a Safe Robust policy improvement over **any** reference policy with an optimal statistical rate. Following the approaches in ATAC can only guarantee a sub-optimal rate, and it is highly non-trivial to extend their results to the constrained setting.
>
> - **Response to compare with baselines:** We thank the reviewer for pointing out the papers CDT and FISOR. The reason we did not incorporate these two algorithms as additional baselines is that their setups do not align well with ours. CDT introduces additional information, target reward, and target cost, during evaluation. These pieces of information are not required in the evaluation process of WSAC and the other baselines we have chosen. Additionally, FISOR considers a different setting from ours as they consider a different type of constraint which is defined for any possible state. For the addition simulations with different cost limits, we report the average performance of our results and other baselines in the following Table (Table 1) under cost limits [10,20,40] for BallCircle and CarCircle and [20, 40, 80] for PointButton and PointPush following the standard setup ([R1]). We can observe that WSAC maintains safety across **all** environments, and WSAC's performance is comparable to or even better than the best baseline in each environment. We will add the details in the final revision.
>
> |   | BC Reward $\uparrow$ | BC Cost $\downarrow$ | Safe-BC Reward $\uparrow$ | Safe-BC Cost $\downarrow$ | CDT Reward $\uparrow$ | CDT Cost $\downarrow$ | BCQL Reward $\uparrow$ | BCQL Cost $\downarrow$ | BEARL Reward $\uparrow$ | BEARL Cost $\downarrow$ | CPQ Reward $\uparrow$ | CPQ Cost $\downarrow$ | COptiDICE Reward $\uparrow$ | COptiDICE Cost $\downarrow$ | WSAC Reward $\uparrow$ | WSAC Cost $\downarrow$ |
> |-------------|----------------------|----------------------|---------------------------|---------------------------|-----------------------|-----------------------|------------------------|------------------------|-------------------------|-------------------------|-----------------------|-----------------------|---------------------------|---------------------------|-----------------------|-----------------------|
> | BallCircle  | 0.74                 | 4.71                 | 0.52                      | 0.65                      | 0.77                  | 1.07                  | 0.69                   | 2.36                   | 0.86                    | 3.09                    | 0.64                  | 0.76                  | 0.70                      | 2.61                      | **0.74**                  | **0.51**                  |
> | CarCircle   | 0.58                 | 3.74                 | 0.50                      | 0.84                      | **0.75**                  | **0.95**                  | 0.63                   | 1.89                   | 0.74                    | 2.18                    | 0.71                  | 0.33                  | 0.49                      | 3.14                      | 0.65                  | 0.55                  |
> | PointButton | 0.27                 | 2.02                 | 0.16                      | 1.10                      | 0.46                  | 1.57                  | 0.40                   | 2.66                   | 0.43                    | 2.47                    | 0.58                  | 4.30                  | 0.15                      | 1.51                      | **0.11**                  | **0.55**                  |
> | PointPush   | 0.18                 | 0.91                 | 0.11                      | 0.80                      | 0.21                  | 0.65                  | **0.23**                   | **0.99**                   | 0.16                    | 0.89                    | 0.11                  | 1.04                  | 0.02                      | 1.18                      | 0.07                  | 0.61                  |
>
>
> **Table 1: The normalized reward and cost of WSAC and other baselines for different cost limits. Each value is averaged over 3 distinct cost limits, 20 evaluation episodes, and 3 random seeds.**
>
> [R1] uxin Liu, Zijian Guo, Haohong Lin, Yihang Yao, Jiacheng Zhu, Zhepeng Cen, Hanjiang Hu, Wenhao Yu,114
> Tingnan Zhang, Jie Tan, et al. "Datasets and benchmarks for offline safe reinforcement learning". arXiv preprint115
> arXiv:2306.09303, 2023.116

---

> > ### Comment · Reviewer_HDvF · 2024-08-11
> >
> > Thank you for your response. The evaluation of safety should focus on whether the algorithm can still ensure safety under a single cost limit. Averaging the number of safety constraint violations across multiple cost limits does not accurately represent the policy's safety, as exceeding the cost limit in practical applications can result in unsafe outcomes. It would be better if the author could separately present the model's performance under different cost limits to demonstrate the safety guarantees provided by their theoretical approach. Furthermore, the author has selected only a very limited subset of environments in OSRL for comparison (4 out of 38 environments), which does not sufficiently demonstrate the algorithm's advantages over the baseline. Based on these points, I maintain my current score.

---

> ### Author Response · Authors · 2024-08-11
>
> > Averaging the number of safety constraint violations
>
> **There seems to be some confusion.** In our paper, we used a single cost limit (thus, there was no point in taking an average) similar to what the baselines (CDT, COptiDICE, CPQ) did in their original papers. The average is taken over different random seeds, but we used  **a single cost limit**. We believe that the results over different random seeds in Table 2 can correctly reflect the true performance of our approach. From Table 2, it is clear for the single cost limit, our approach is the *only one* that can satisfy the constraints across different benchmarks.
>
> During the rebuttal phase, per the reviewer's request, we ran the algorithm with different cost limits following the exact format used in the Offline Safe RL Benchmark (OSRL) paper, where they report the average across performance with different cost limits. It is clear that our algorithm performs well, as no other algorithms are consistently safe, in terms of average performance. To further address the reviewer’s concern, we report individual results with different cost limits under our algorithm in the following table, where our algorithm achieves very low costs and high safety rates and is nearly safe for all environments. Note that due to time constraints in the rebuttal phase, we did not perform any parameter tuning, and we believe we can further improve the performance (both in terms of reward and safety) if we do so. Note that, we can't compare with other baselines since the OSRL paper doesn't have the results for different cost limits (they only have the average results). We also believe that although one of our main contributions is [theoretical](https://openreview.net/forum?id=82Ndsr4OS6&noteId=fBIu70VHUw), our approach with theoretical support is quite general and has the potential to be incorporated into other practical Safe-RL algorithms.
>
> | | Reward ↑ | Cost ↓ | Reward ↑ | Cost ↓ | Reward ↑ | Cost ↓ |
> |-------------|----------|--------|----------|--------|----------|--------|
> |     Cost Limit          | 10       |        | 20       |        | 40       |        |
> | BallCircle  | 0.71     | 0.10   | 0.76     | 1.17   | 0.75     | 0.27   |
> | CarCircle   | 0.60     | 0.07   | 0.67     | 0.99   | 0.68     | 0.59   |
>
> |  | Reward ↑ | Cost ↓ | Reward ↑ | Cost ↓ | Reward ↑ | Cost ↓ |
> |-------------|----------|--------|----------|--------|----------|--------|
> |      Cost Limit        | 20       |        | 40       |        | 80       |        |
> | PointButton | 0.01     | 0.47   | 0.13     | 0.67   | 0.18     | 0.51   |
> | PointPush   | 0.10     | 1.11   | 0.07     | 0.52   | 0.05     | 0.21   |
>
> **Table 1: The normalized reward and cost of WSAC for different cost limits.  Each value is averaged over 20 evaluation episodes, and 3 random seeds.**
>
> >  limited subset of environments
>
> We focus on environments where most (if not all) baselines are unsafe (e.g., no baselines are safe in PointButton, and only BCQL is safe in PointPush). Yet, we show that our proposed approach can achieve safety while maintaining good reward which points towards its efficacy.  We believe these environments are both challenging and representative, effectively justifying an algorithm's ability to guarantee safety.
>
> We would like to emphasize that our main contribution in this paper is to address the Safe Robust Policy Iteration (SRPI) and policy coverage limitations in theoretical offline safe RL which are quite important in the theoretical safe-RL community (see Table 1). For example, none of the existing approaches (including the baselines) have a **safe robust policy improvement guarantee** using only a *single policy coverage assumption*. In particular, our approach provides a way to achieve a safe policy using only offline data with bare minimum richness (single policy coverability). Existing approaches that are based on primal-dual concept require more richness in data (all policy concentrability which is not possible to achieve in practice). Please see Table 1 and discussion in Introduction. Furthermore, the baselines do not provide any theoretical guarantees, since they aim to design practical algorithms. To demonstrate the empirical efficiency of our approach, we included four challenging environments in our paper. We agree that running the algorithm on all 38 environments in the baseline Safe RL paper would surely have values, and we will try to evaluate our approach on more baselines in the final version.  However, we believe that the environments we selected are sufficient to demonstrate the core ideas of our paper and validate the theoretical insights. Even the state-of-the-art algorithms  (without theoretical guarantees) only include a limited number of representative environments in their papers; for instance, CDT has 5, CPQ has 3, and COptiDICE has 4.
>
> We sincerely hope that the reviewer will reevaluate the rating based on the novel contributions of our paper.

---

> > ### Comment · Reviewer_HDvF · 2024-08-12
> >
> > After considering the positive feedback provided by other reviewers on the theoretical aspects, I will increase the score from 3 to 5. However, I still think that the main proof core and the concept of safe policy improvement primarily originate from ATAC. I suggest that other reviewers might want to further compare the theoretical aspects of ATAC with those presented in this paper.
> >
> > Cheng, Ching-An, et al. "Adversarially trained actor critic for offline reinforcement learning." International Conference on Machine Learning. PMLR, 2022.
> >
> > Additionally, concerning the algorithm's performance on benchmarks and the selection of baselines, I think it would be prudent to include more advanced state-of-the-art approaches. After all, CPQ and Copitidice are articles from 2022, and bcql and bearl come from earlier offline RL algorithms. Regarding the recently well-performing algorithms like CDT and FISOR, it would be beneficial for the author to explore or discuss the feasibility of transitioning the WSAC theoretical framework to SOTA.
> >
> > Regarding the issue of non-comparison due to different settings mentioned by the author, it's noteworthy that while CDT requires the introduction of additional information such as target reward and target cost, the baseline in the paper, including WSAC, also necessitates a human-defined cost limit. For FISOR, since safety is a key goal in safe RL research, it doesn't make sense to say that stricter safety constraints stop us from comparing safety between different algorithms.

---

> ### Author Response · Authors · 2024-08-13
>
> Thanks for increasing your score and engaging with us.
>
> > Technical Differences with ATAC
> ATAC is the first paper in the literature to investigate the property of RPI. While we certainly draw insights from their results, our work has the following significant differences.
>
> - First, we focus on the **constrained Markov decision process (CMDP)** setup rather than an unconstrained setup. The CMDP setup is fundamentally different from the unconstrained one. For example, in CMDP, the optimal policy can be stochastic, unlike in the unconstrained MDP. In the constrained setup, it is essential to bound both the sub-optimality of the reward and the constraint violation, whereas, in the unconstrained setup, only the sub-optimality of the reward needs to be bounded. Naturally, the analytical results and algorithms differ significantly from those in the unconstrained ATAC setup.
>
> - Furthermore, our approach to training the critics is different from that of ATAC. ATAC uses a squared Bellman error, while we utilize the average Bellman error to train the critic. Consequently, we achieve a $1/\sqrt{N}$ sample complexity error, while ATAC achieves a $1/N^{1/3}$ sample complexity error. The key difference is that we use an importance-weighted Bellman error to obtain an unbiased estimator of the critic for both the reward and cost, tuning the weight parameter to achieve a better rate, unlike ATAC.
>
> - Moreover, while primal-dual-based methods exist for solving the offline CMDP, achieving robust safe policy improvement and relaxing the assumption of all-policy concentrability remained open challenges (Table 1 in our paper). We resolved this open question. Our approach guarantees robust policy improvement, so if a safe reference policy (e.g., a safe behavioral policy) is provided, our algorithm will yield a policy that remains safe without sacrificing reward. Such a guarantee was previously missing from the literature. As pointed out in the introduction, **all-policy** concentrability is difficult to satisfy in practice, especially in a safe setting where the dataset may not cover state-action pairs from an unsafe policy. Instead, we only require single-policy concentrability, making our theoretical results highly impactful for the safe RL community.
>
> - We consider a policy improvement over **any reference policy** not only the behavior policy.
>
> - It is worth noting that to provide such a guarantee, we developed a rectified penalty-based approach rather than a primal-dual-based one. As a result, our analysis differs from existing primal-dual approaches. In fact, the existing primal-dual-based approaches only guarantee all-policy concentrability, so our analytical insights and proposed approach open new avenues for finding a safe policy from an offline dataset.
>
> > compare with the SOTA baselines
>
> it is crucial to compare with the state-of-the-art (SOTA) baselines to demonstrate the strength of our proposed approach. Therefore, we compare our practical version with existing approaches on selected benchmarks. As requested by the reviewer, we have included results from CDT in our rebuttal. We apologize for not making this clearer earlier. Notably, ours is the only approach that achieves safety, underscoring the efficacy of our method. Additionally, CDT uses a transformer architecture, which naturally results in a longer computational time compared to our [approach](https://openreview.net/forum?id=82Ndsr4OS6&noteId=VR6S2gv84s).
>
>  Finally, we would like to mention that in safe RL, there are two types of constraints: soft constraints (in the long-term average sense) and hard constraints (step-wise). It is difficult to say which one is more important because, in the long-term average case, taking some risk is necessary; otherwise, the problem would be no different from an unconstrained problem. Moreover, the existing solutions in theoretical safe RL for addressing these two types of constraints are significantly different. We agree that the cost limits are chosen by humans, and we appreciate the reviewer pointing out that a fairer comparison should consider different sets of cost limits. We observe that there is a trend: the reward increases when the cost limit is higher. To understand the differences between two settings, we can also observe that as reported in the FISOR paper, some environments (e.g., SwimmerVel, CarButton1, CarGoal2) exhibit very low or even negative rewards because they aim to learn very safe policies. We will definitely add more discussion in the final revision.
>
> We are also happy to learn more about the reviewer's opinion on selecting the cost limits. What we typically do is to make sure the problem itself is feasible and the optimal solution is stochastic in synthetic CMDPs and follow what people use (like OSRL and other baselines) in complicated environments.
>
> **We hope that our response addresses the concerns of the reviewer and is open to further discussion. Thank you again for raising the score!**

---

### Official Review · Reviewer_Tiai · 2024-07-13

**Soundness:** 3
**Presentation:** 3
**Contribution:** 3
**Rating:** 7
**Confidence:** 3

**Summary:**

For safe RL methods, a desired property is Safe Robust Policy Improvement(SRPI), which means the learned policy is always at least as good and safe as the baseline behavior policies. But it's not achieved yet.

Also, the traditional Actor-Critic framework may suffer from insufficient data coverage, which may fail to provide an accurate estimation of the policy for unseen states and actions. To address the issue, [45] and [11] use absolute pessimism or relative pessimism. However, his kind of approach fails to achieve the optimal statistical rate of $\sqrt{N}$. For addressing efficient policy improvement, the most commonly used approach for addressing safe RL problems is primal-dual optimization, but this method requires all policy concentrability, that is, the dataset must cover all possible strategies, which is impractical for the safe-related dataset.

In contrast, the authors propose an aggression-limited objective function,  the high-level intuition behind it is that by appropriately selecting a 𝜆,  all unsafe policies are penalized.  As a result, the policy that maximizes the objective function is the optimal safe policy. This formulation is fundamentally different from the traditional primal-dual approach as it does not require dual-variable tuning, and thus, does not require all policy concentrability.

Beyond that, the proposed method also proved to achieve SRPI.

[45] Tengyang Xie, Ching-An Cheng, Nan Jiang, Paul Mineiro, and Alekh Agarwal. Bellman-consistent pessimism for offline reinforcement learning.

[11]Ching-An Cheng, Tengyang Xie, Nan Jiang, and Alekh Agarwal. Adversarially trained actor critic for offline reinforcement learning

**Strengths:**

1.  The writing logic is clear and reasonable. This paper is written with solid insights and emphasizes research gaps and innovations.

2. The authors conducted wide-range experiments and comparisons with other methods. And in terms of safety, the proposed method achieved SOTA performance.

3. There are no obvious red flags or drawbacks in this paper.

**Weaknesses:**

1. In this paper's setting, safety is measured purely by cost, which is not always practical. e.g. in the real world, the cost function could be implicit or impossible to get.

2. The authors could try to combine the proposed method with other RL methods for a certain application to further justify its effectiveness.

**Questions:**

The paper looks good. And there is a suggestion for future directions.

Bridge the gap between theory and practice. Current RL methods have some bottlenecks like long-horizon tasks, safety, sample efficiency, etc.. Most of the methods to solve these bottlenecks come out of intuition instead of theory deduction. And some methods leverage external tools like foundation models(e.g. LLM), and control theories. It might be interesting to explain why these intuition or external tools work in view of theories.

**Limitations:**

The authors did not provide limitations.

A possible limitation may come from the experiment on PointPush, in which the simple BC's reward outperformance proposed method without sacrificing too much safety.

And as stated in weakness, safety is expressed purely by the cost function, which sometimes is hard to get in the real world.

---

> ### Author Rebuttal · Authors · 2024-08-07
>
> We greatly appreciate the reviewers’ positive evaluation of the novelty of this paper. The current formulation can be applied to the case with $0-1$ cost, indicating whether a constraint is violated or not at each step. Nevertheless, if we only get feedback over the entire trajectory whether it is safe or not rather per step feedback, it is an indeed interesting future research direction on how to address such a scenario. In fact, even in the online setting such a scenario is yet to be resolved. We add a discussion.
>
>  We strongly agree with the reviewer on the gap between theories and practical algorithms in RL and safe RL. Our approach is motivated to take a step towards bridging the gap as we consider offline setup and we only need single policy coverability for the theoretical results. Regarding the comments on external tools like LLMs, one possible direction could be to consider the cost preference (like RLHF) instead of cost function observations and determine whether it is possible to design an algorithm with provable guarantees of safety.

---

> > ### Comment · Reviewer_Tiai · 2024-08-12
> >
> > Thank you for the response! And it helps me keep a positive opinion of this work, so I would maintain my score.

---

> > > ### Author Response · Authors · 2024-08-13
> > >
> > > Thank you very much for your acknowledgment and your positive feedback on our work!

---

### Official Review · Reviewer_U4TL · 2024-07-13

**Soundness:** 3
**Presentation:** 3
**Contribution:** 3
**Rating:** 6
**Confidence:** 3

**Summary:**

This paper proposes weighted safe actor-critic, and provides corresponding theoretical analysis on its optimal statistical rate. Some interesting technical tools were introduced. The authors also implement a practical version of WSAC and evaluate it against SOTA offline safe RL baselines in continuous control tasks.

**Strengths:**

(1) This paper addresses offline safe RL with adversarial trained weighted AC framework, showing its optimal statistical convergence rate.

(2) Under the perfect function approximation assumption, the authors show WSAC outperforms any reference policy while maintaining the same level of safety. Besides, the theoretical finding on safe robust policy improvements bring insight to the offline safe RL methods.

(3) Empirically, the authors provide a comparison to a set of baselines in OSRL benchmarks.

**Weaknesses:**

See more discussion in the question parts.

**Questions:**

(1) **Clarification of "Adversarial"**: I'd like to have a clarification of the term "adversarial" in this offline safe RL problem. Do the authors mean "adversarial" in that the cost critic always update cost critic via optimism and reward critic via pessimism? Since there are other formats of adversarial robustness in other components of safe RL [1], I may be helpful if the authors could clarify it in the early stage of this paper.

(2) **Finite selection of W**: the current WSAC algorithm prototype only considers a discrete selection of $w$, can they be arbitrarily assigned for different offline datasets and environments? Intuitively if an arbitrary $w$ is close enough to its neighbor $w$ in set $\mathcal{W}$, we can still provide analytical bound of performance under slightly different $w$.

(3) **Lack of discussion of assumption gap in the pratical version**: the authors may describe how certain assumptions may not hold in practice, as they are some seemingly relatively strong assumptions in the theoretical analysis, like approximate realizability, though some of them have already got loosened. Also, some experiment details (e.g. the behavior policy or the oracle policy) can be discussed in the appendix to provide more contexts in how WSAC can help practically.

(4) **Capability under sparse-cost setting**:  in many real-world applications, safety violations will occur only in long-tail events. Can the reweighting scheme also address such long-tail cases in the cost critic learning?  Can the current WSAC framework can be extended for this kind of analysis based on the weighting technique?

(5) **Extension of the current framework to multi-constraint settings**: in real-world applications, there might be multiple objectives and constraints, can the WSAC framework adapt to similar settings?

(6) **Selection of weight W in the experiments**: Compared to algorithm 1, the practical implementation of WSAC seems to miss $\mathcal{W}$, how is this importance weight computed in practice?


> [1] Liu, Zuxin, et al. "On the robustness of safe reinforcement learning under observational perturbations." *arXiv preprint arXiv:2205.14691* (2022).

**Limitations:**

The authors have clearly defined the scope and discussed the limitations of this paper.

---

> ### Author Rebuttal · Authors · 2024-08-07
>
> Thank you for your positive feedback on our paper! We appreciate your support and comments. Please find our point-by-point response to your questions below.
>
> - **Response to clarification of "Adversarial":** The adversarial training in this paper is designed based on the concept of relative pessimism [11,4,53], which aims to optimize the worst-case relative performance over uncertainty. In particular, we adversarially train the critic to find weighted Bellman-consistent scenarios where the actor is inferior to the reference/behavior policy (Eq. (2) and Eq. (4)). Note that we consider an offline setup, hence, we cannot access any new data, rather whatever data is available, we have to find policy based on that. Without adversarial training, if some state-action pairs are not covered in the dataset, one can set (incorrectly) a very high $ Q$ value without affecting the training loss. The adversarially trained critic addresses this issue by setting the lowest possible $Q$-value avoiding setting a high value for our-of-distribution values. We will clarify and discuss the differences between our approach and the paper mentioned by the reviewer in the final version.
>
> - **Response to finite selection of $W$:** The selection of $W$ is not arbitrary: it needs to be chosen by maximizing the importance-weighted average Bellman error (Eq. (2) and Eq. (4)). This is crucial in the formulation because maximization over $w$ in the importance-weighted average Bellman regularizer ensures that the Bellman error is small when averaged over measure $\mu \cdot w$ for any $w \in W$. This can control the suboptimality of the learned policy and guarantee the optimal statistical rate of $1/\sqrt{N}.$
>
> - **Response to lack of discussion of assumption gap in the practical version:**  For the assumptions made in our paper, as far as we know, we require minimal assumptions in offline RL (as shown in Table (1)), especially in safe offline RL. The single-policy coverage required by our approach is far milder than the all-policy coverage assumption as it is impractical to assume that the dataset would contain state-action pairs from unsafe policies. Thus, we achieve our result for a more practical and realistic set of assumptions (and it matches the same set of assumptions for the unconstrained case). For the practical purpose, we think the most concerning part is that approximate realizability in Section 3.2 may not hold in practice. However, in our experience, as we usually use neural networks to approximate the function class $F$, it is likely that the approximate realizability error is small as long as we have a sufficiently rich neural network. The behavior policy is easy to achieve, even if it is not given to us, since extracting the behavior policy from an offline dataset is not difficult with behavior cloning (BC). In particular, we can estimate the learned behavior policy $\hat{\pi}\_\mu$ as follows: $\forall s \in D, \hat{\pi}\_\mu(a \vert s) \leftarrow \frac{n(s,a)}{n(s)}$, and $\forall s \notin D, \hat{\pi}\_\mu(a \vert s) \leftarrow \frac{1}{\vert A \vert}$, where $n(s,a)$ is the number of times $(s,a)$ appears in the offline dataset $D$. Essentially, the estimated BC policy matches the empirical behavior policy on states in the offline dataset and takes uniform random actions outside the support of the dataset. It is easy to show that the gap between the learned policy $\hat{\pi}\_\mu$ and the behavior policy $\pi\_\mu$ is upper bounded by $\min\lbrace 1, \vert S \vert / N \rbrace$ [R1, R2]. We can have a very accurate estimate as long as the size of the dataset is large enough. We will add more details in the revision.
>
> [R1]: Kumar, Aviral, Joey Hong, Anikait Singh, and Sergey Levine. "Should i run offline reinforcement learning or behavioral cloning?." In International Conference on Learning Representations. 2021.
>
> [R2]: Rajaraman, Nived, Lin Yang, Jiantao Jiao, and Kannan Ramchandran. "Toward the fundamental limits of imitation learning." Advances in Neural Information Processing Systems 33 (2020): 2914-2924.
>
>
> - **Response to capability under sparse-cost setting:**  Maximizing the importance-weighted average Bellman regularizer aims to control the Bellman error. We believe this can also be applied to the long-tail constraint case. Our results can be generalized to a high probability bound (might be loose) using Markov's inequality, and it is possible to show some results under the CVaR objective. Of course, rigorous proofs and experiments are needed. We will explore these interesting directions in the future.
>
>
> - **Response to extension of the current framework to multi-constraint settings:**  Yes, the current approach is readily to generate to the case with multiple constraints by simply considering multiple constraints in the objective function: $f_r^k(s,a) - \sum\_{i=1}^I \lambda\_i \lbrace f_{i,c}^k(s,a) - f_{i,c}^k(s.\pi_{ref}) \rbrace\_+,$ where $f\_{i,c}^k$ is the $i$th constraint. We only have to tune $\lambda_i$, and $\beta_{c,i}$ values corresponding to constraint $i$. We will add some discussions in the revision.
>
>
> - **Response to the selection of weight W in the experiments:** In experiments, we take $\mathcal{W} = \{0, C_\infty\}$ for computation effectiveness. Then we can reduce $\epsilon \_{D}(\pi, f)$ (Eq. (4)) to $C\_{\infty}E\_{D}[(f(s,a)-r-\gamma f(s',\pi))^2]$. In the environment, we simply set $C_\infty$ to be 1 in the neural case and 5 in the tabular setting.

---

> > ### Comment · Reviewer_U4TL · 2024-08-11
> >
> > I thank the authors for their detailed response, especially their justification of the assumption. One clarification question on the statement: " The behavior policy is easy to achieve, even if it is not given to us, since extracting the behavior policy from an offline dataset is not difficult with behavior cloning (BC)." Since the example you provided is using pure tabular case, which is even weaker than your assumption (discrete action space + complex state space). How do you justify the difficulty of behavior policy extraction given the safety constraints in offline safe RL?
> >
> > Another follow-up question about the gap between theoretical assumptions and their empirical practicability is that most OSRL baselines have continuous action and continuous state space, and the authors propose a practical version of WSAC in the appendix. I'm curious about whether the authors can provide some insights on how the current theoretical guarantees can be extended to the non-tabular cases, if applicable.
> >
> > Besides the above two questions, most of the other questions are well-addressed by the author's response. I thank the authors again for their dedicated efforts and will determine my final score after this round of discussion.

---

> ### Author Response · Authors · 2024-08-12
>
> We greatly thank the reviewer for considering reevaluating our paper and for raising these two interesting questions.
>
> >Extracting the Behavior Policy
>
> In the standard offline RL setting, we assume that the dataset is generated by some behavior policy $\mu(s,a)$ and is i.i.d. Therefore, as long as the state-action space is finite, the method we provided guarantees that we can accurately estimate the behavior policy. This holds true whether we are considering safe RL or regular RL, as the only difference is the additional information regarding the cost function, while the process of generating the dataset remains the same.
>
> For more complicated cases, such as when the state space is continuous, we can use a neural network to approximate the policy by minimizing the distance between the learned policy and the behavior policy. Alternatively, we could use DAgger (Dataset Aggregation) to achieve an even better policy. However, in such cases, the assumptions made in offline RL (not only in our paper but in the field generally) may no longer hold, particularly the data/policy coverage assumption. This is why we argue in our paper that single-policy coverage is crucial since it is much weaker than the full-policy coverage assumption.
>
> >extended to the non-tabular cases
>
>  There seems to be some confusion. We consider the function approximation setting, not a tabular setting, and the theoretical results are independent of the size of the state and action space under the given assumptions. The practical version of our algorithm is designed to develop a deep neural network approach for more complex environments. In order to handle the continuous state and action spaces, we use an actor-critic approach to solve the optimization problem (2), which aligns with the objective in our theoretical version. Specifically, we can use the aggression-limited objective to train the actor-network, which is feasible by considering two Q-value neural networks (one for reward and one for cost). The method used in our practical version for the weighted Bellman regularizer is a very simplified version. However, we believe it is possible to approximate $w(s,a)$ with another neural network such that the weighted Bellman error is minimized when the critic network is fixed. We will add more discussions and possibly some results in the revision.
>
> Please let us know if you have further questions and comments, we are glad to have more discussions.

---

> > ### Comment · Reviewer_U4TL · 2024-08-13
> >
> > I thank the authors for their extensive replies. Also, I read the discussion between the authors and reviewer HDvF. I think most of my questions have been well-addressed with the clarification of their theoretical contribution and the new empirical evidence. I will raise my score to 6 in favor of the acceptance.

---

> > > ### Author Response · Authors · 2024-08-13
> > >
> > > Thank you very much again for your great comments and for taking the time to engage with us. We sincerely appreciate your acknowledgment and positive feedback on our work!

---

### Official Review · Reviewer_1bfE · 2024-07-29

**Soundness:** 3
**Presentation:** 3
**Contribution:** 3
**Rating:** 6
**Confidence:** 3

**Summary:**

This paper introduces a principled approach for safe offline reinforcement learning (RL), aimed at robustly optimizing policies beyond a given reference policy, particularly when constrained by the limited data coverage of offline datasets.
The traditional constrained actor-critic methods face challenges including (1) coping with insufficient data coverage, (2) ensuring robust policy improvement, and (3) facilitating computationally efficient actor optimization.
To address these limitations, this study presents the Weighted Safe Actor-Critic (WSAC) framework. WSAC incorporates (1) a pessimistic bias through a weighted average Bellman error, (2) theoretical assurances for robust policy improvement, and (3) an efficiency advantage over traditional primal-dual optimization methods.
WSAC employs an aggression-limited objective function, which discourages unsafe policies, relying on less stringent assumptions compared to prior methodologies. Furthermore, WSAC leverages the reference policy as a no-regret policy optimization oracle, allowing for safe policy training.
The efficacy of WSAC is demonstrated across four benchmark environments: BallCircle, CarCircle, PointButton, and PointPush. The results indicate that WSAC effectively optimizes policies to maximize cumulative rewards while maintaining cumulative costs below predefined thresholds.

**Strengths:**

This work is well-grounded in rigorous theoretical principles that effectively support the proposed WSAC method.
The method's foundation on pessimistic value estimation and robust policy improvement is both mathematically sound and appropriate for the challenges of safe offline RL.
Furthermore, the integration of an adversarial training component within the actor-critic architecture introduces a novel strategy for mitigating common issues such as insufficient data coverage in offline RL.
This significantly bolsters the robustness of the resulting policies against shifts in data distribution, a crucial factor for applications in real-world scenarios.
By addressing the critical issue of safety in policy optimization with a computationally efficient approach, I believe that the paper makes a substantial contribution to moving the field towards practical, deployable reinforcement learning systems capable of addressing real-world challenges.

**Weaknesses:**

The proposed Weighted Safe Actor-Critic (WSAC) method in this submission is contingent upon the availability of an explicit reference policy, such as a behavior policy derived from the offline dataset.
This requirement could make training difficult in scenarios where extracting a reliable reference policy from the offline data is challenging, particularly for algorithms aiming to be behavior-agnostic in offline RL settings.

Additionally, the authors claim in line 239 that "Our approach is very computationally efficient and tractable compared with existing approaches." However, the absence of empirical evidence, such as wall-clock time comparisons, to substantiate this claim weakens their argument.
Providing such comparative data would significantly strengthen their case for computational efficiency.

Moreover, the paper does not include ablation studies to elucidate the contributions of the three key components of WSAC: (1) weighted Bellman error, (2) aggression-limited objective, and (3) no-regret policy optimization using a single reference policy. Identifying which of these components is most critical to performance enhancement would provide clearer insights into the framework's effectiveness and areas for potential improvement.

**Questions:**

Q1.  Could you elaborate on the sensitivity of the hyperparameters such as $\beta_c$, $\beta_r$, $\lambda$? Understanding their influence on the model's performance and robustness would be beneficial, especially in varying training conditions.

(Minor Comments)

1. In obj 2, the Bellman error coefficients $\beta$ used in the reward and cost constraints appear to have different values, indicated by $\beta_c, \beta_r$. To avoid confusion, it would be prudent to denote these coefficients separately throughout the manuscript to reflect their distinct roles and values.

2. The typo in line 231 : "WSAC sovles" →  "WSAC solves."

**Limitations:**

The authors addressed their limitations in Section 4 and Conclusion and the broader societal impact in Checklist #10: Broader Impacts.

---

> ### Author Rebuttal · Authors · 2024-08-07
>
> Thank you for your positive feedback on our paper! We appreciate your support and comments. Please find our point-by-point response to your questions below.
>
> -  **Response to the reference policy:** We would like to mention that extracting the behavior policy from an offline dataset is not difficult with behavior cloning (BC). In particular, we can estimate the learned behavior policy $\hat{\pi}\_\mu$ as follows: $\forall s \in D, \hat{\pi}\_\mu(a \vert s) \leftarrow \frac{n(s,a)}{n(s)}$, and $ \forall s \notin D, \hat{\pi}\_\mu(a \vert s) \leftarrow \frac{1}{\vert A \vert} $, where $n(s,a)$ is the number of times $(s,a)$ appears in the offline dataset $D$. Essentially, the estimated BC policy matches the empirical behavior policy on states in the offline dataset and takes uniform random actions outside the support of the dataset. It is easy to show that the gap between the learned policy $\hat{\pi}\_\mu$ and the behavior policy $\pi_\mu$ is upper bounded by $ \min \lbrace1, \vert S \vert / N \rbrace$ ([R1, R2]). We can have a very accurate estimate as long as the size of the dataset is large enough. In addition, in many applications such as networking, scheduling, and control problems, there are existing good enough reference policies. In these cases, a safe robust policy improvement over these reference policies has practical value.
>
>
> [R1]: Kumar, Aviral, Joey Hong, Anikait Singh, and Sergey Levine. "Should i run offline reinforcement learning or behavioral cloning?." In International Conference on Learning Representations. 2021.
>
> [R2]: Rajaraman, Nived, Lin Yang, Jiantao Jiao, and Kannan Ramchandran. "Toward the fundamental limits of imitation learning." Advances in Neural Information Processing Systems 33 (2020): 2914-2924.
>
> - **Response to computationally efficient:** We thank the reviewer for pointing out our statements on computational efficiency compared to existing approaches. We claim our approach is *efficient* and *tractable* compared to [19, 26] mainly because: [26] requires two FQI inner loops for policy improvement and three additional inner loops for policy evaluations, while [19] also requires an inner loop for offline policy evaluation. However, our algorithm does not have any inner loop for extra OPE. To demonstrate the efficiency, in the following Table (Table 1), we report the training times of our method and other baselines for the Car Goal environment using one NVIDIA GeForce RTX 3080 Ti. We observe that our practical version still has a faster training time, which is very time-efficient compared to others. We will make this clear in the revision.
>
> |         | BEARL | CPQ | CDT | COptiDICE | WSAC |
> |:---------|:-------:|:-----:|:-----:|:------------:|:------:|
> | Time    |  121  | 118 | 465 | 117        | 115  |
>
> **Table 1: Training Time (seconds) for 200 steps**
>
> - **Response to ablation studies:**  The weighted Bellman regularize, aggression-limited objective, and no-regret policy optimization together guarantee our theoretical results. We did an ablation study in the tabular setting and the results can be found in the following table (Table 2). The results of the ablation study indicate that the weighted Bellman regularization and no-regret policy optimization ensure the safety of the algorithm, while the aggression-limited objective ensures the algorithm to achieve higher rewards without compromising safety.
> | Components                                                   | Cost  | Reward |
> |--------------------------------------------------------------|-------|--------|
> | ALL                                                          | 0.016 | 0.766  |
> | W/O  no-regret policy optimization    | 0.016 | 0.766  |
> | W/O  Aggression-limited objective  | 0.016 | 0.765  |
> | W/O   Weighted Bellman regularize | 0.181 | 0.624  |
>
> **Table 2: Ablation study under tabular case (cost limit  = 0.1)**
>
>  - **Response to hyperparameter sensitivity of $\beta$ :**  Note that our main result remains the same for SRPI as long as $\beta \geq 0$, indicating that our approach is highly robust across a wide range of $\beta$ values. We use different $\beta_r$ and $\beta_c$ in the practical version because it allows us to easily make a trade-off between rewards and costs, as different environments have varying sensitivities to rewards and costs. To address the reviewer's comments about hyperameter sensitivity, we provide the rewards and costs under different sets of $\beta_r=\beta_c\in \{1,0.5,0.05 \}$ and $\lambda\in\{[0,1],[0,2],[1,2]\}$ (since our $\lambda$ only increases, the closed interval here represents the initial value and the upper bound of $\lambda$) to demonstrate the robustness of our approach in the tabular setting in Figure 1 (Please see the uploaded file). We can observe that the performance is almost the same under different sets of parameters and different qualities of behavior policies. We will add more details in the revision.

---

> > ### Comment · Reviewer_1bfE · 2024-08-11
> >
> > Thank you for addressing my concerns. It would be beneficial if the authors could report the error bars (e.g., standard errors or confidence intervals) of the computational cost and results of the ablation study in the revision.
> >
> > I still believe this paper makes sufficient contributions to the offline RL community, so I will be maintaining my rating.

---

> > > ### Author Response · Authors · 2024-08-12
> > >
> > > We thank the reviewer again for the positive evaluation of our paper. We provide the error bars in the following tables.
> > >
> > > |                    | BEARL  | CPQ    | CDT    | COptiDICE | WSAC   |
> > > |--------------------|--------|--------|--------|-----------|--------|
> > > | **Time (seconds)** | 120.0  | 113.8  | 464.6  | 112.0     | 116.40 |
> > > | **STD**            | 1.41   | 2.13   | 1.62   | 5.05      | 1.85   |
> > > | **Confidence Interval** | (116.07, 123.93) | (107.88, 119.37) | (460.09, 469.11) | (97.95, 126.05) | (111.25, 121.55) |
> > >
> > > **Table 1:** Training Time (seconds) for 200 steps over 5 repeat experiments
> > >
> > >
> > > | Components                                                        | Cost  | Reward | Cost STD | Reward STD | Cost Interval    | Reward Interval    |
> > > |-------------------------------------------------------------------|-------|--------|----------|------------|------------------|--------------------|
> > > | **ALL**                                                           | 0.014 | 0.788  | 0.006    | 0.004      | (0.00, 0.03)     | (0.78, 0.80)       |
> > > | **W/O no-regret policy optimization**   | 0.014 | 0.788  | 0.006    | 0.004      | (0.000, 0.028)   | (0.779, 0.798)     |
> > > | **W/O Aggression-limited objective**  | 0.014 | 0.788  | 0.006    | 0.005      | (0.000, 0.028)   | (0.778, 0.798)     |
> > > | **W/O Weighted Bellman regularizer**  | 0.323 | 0.684  | 0.061    | 0.017      | (0.185, 0.462)   | (0.645, 0.724)     |
> > >
> > > **Table 2:** Ablation study under tabular case (cost limit is 0.1) over 10 repeat experiments

---

### Author Rebuttal · Authors · 2024-08-07

We would like to thank all the reviewers for their thoughtful evaluations. In this global rebuttal, we point out our main contributions and address the common concerns of the reviewers. We respond to each individual reviewer in each individual rebuttal separately as well.

**Our Contributions**:

* We consider an offline constrained MDP (CMDP) problem. Unlike the unconstrained MDP, here, one needs to learn a policy that maximizes reward while simultaneously satisfying constraint using only offline data. We prove that our algorithm, which uses weighted Bellman error, enjoys an optimal statistical rate of $1/\sqrt{N}$ under partial data coverage assumption.  Note that all the existing approaches for offline safe RL/CMDP (see Table 1) require **all**-policy concentrability which is not possible in practice. In particular, an offline database may not contain state-action pairs covered by every unsafe policy. We achieve our result using a more practical set of assumptions with only single policy concentrability. *This is the first work that achieves such a result using only single-policy $\ell_2$ concentrability.*

* We propose a novel offline safe RL algorithm, called Weighted Safe Actor-Critic (WSAC), which can robustly learn policies that improve upon any behavior policy with controlled relative pessimism. We prove that under standard function approximation assumptions, when the actor incorporates a no-regret policy optimization oracle, WSAC outputs a policy that never degrades the performance of a reference policy (including the behavior policy) for a range of hyperparameters. *This is the first work that provably demonstrates the property of SRPI in offline safe RL setting.*

 *  We point out that primal-dual-based approaches [19 ] must require all-policy concentrability assumption. Thus, unlike, the primal-dual-based approach, we propose a novel rectified penalty-based approach to obtain results using single-policy concentrability. *Thus, we need novel analysis techniques to prove results compared to existing approaches.*

* Furthermore, we provide a practical implementation of WSAC following a two-timescale actor-critic framework using adversarial frameworks similar to [11, 53 ], and test it on several continuous control environments in the offline safe RL benchmark [ 31 ]. WSAC outperforms all other state-of-the-art baselines, validating the property of a safe policy improvement. In particular, from Table 2 (in our paper), it is clear that across all the environments, WSAC is the **only** algorithm that achieves safety and yet has achieved a better or similar reward in most of the environments. Thus, our proposed approach has contributed significantly in both theoretical and practical fronts for safe offline RL.

**New Results**:

* We have now achieved new empirical results. In particular, we evaluate the sensitivity of the hyper-parameters $\lambda, \beta_r, \beta_c$ and observe that our algorithm's performance is robust under different sets of parameters (see the attached Figure).

* We have now conducted an empirical evaluation of the computational efficiency of our proposed approach and observed that it takes a smaller time compared to the state-of-the-art approaches (see Table 1 in the response [here](https://openreview.net/forum?id=82Ndsr4OS6&noteId=T2H496Awkn))

* We have now conducted Ablation studies (see Table 2 in the response [here](https://openreview.net/forum?id=82Ndsr4OS6&noteId=T2H496Awkn))

* We have now compared our approach with another baseline mentioned by the reviewers. We have outperformed the approach. (Please see Table 1 in the response [here](https://openreview.net/forum?id=82Ndsr4OS6&noteId=fBIu70VHUw))

---

### Decision · Program_Chairs · 2024-09-25

**Decision:**

Accept (poster)

**Comment:**

This paper introduces an approach for safe offline reinforcement learning (RL), by introducing Weighted Safe Actor-Critic (WSAC), which incorporates pessimistic bias through a weighted average Bellman error and theoretical guarantees. The efficacy of WSAC is demonstrated across four benchmark environments: BallCircle, CarCircle, PointButton, and PointPush; the results are promising. Although the work seems to be a little bit similar to ATAC, the reviewers overall all recommend acceptance as the work is both theoretically grounded and practically significant.